# Development of a multi-epitope vaccine against *Acinetobacter baumannii*: A comprehensive approach to combating antimicrobial resistance

**Masoumeh Beig[1,2], Mohammad Sholeh[1], Safoura Moradkasani[1], Behzad Shahbazi[3], Farzad Badmasti [1]***

1 Department of Bacteriology, Pasteur Institute of Iran, Tehran, Iran, 2 Student Research Committee, Pasteur Institute of Iran, Tehran, Iran, 3 School of Pharmacy, Semnan University of Medical Sciences, Semnan, Iran

* fbadmasti2008@gmail.com

## Abstract

### Background

The World Health Organization has categorized *Acinetobacter baumannii* (*A. baumannii*) as a critical priority pathogen due to its high antibiotic resistance. This resistance complicates treatment and underscores the urgent need for new antibiotics and strategies. This study developed a multi-epitope vaccine (MEV) to address this significant public health threat.

### Methods

This study employed a computational approach to design MEV targeting *A. baumannii* strain VB7036. Surface-exposed proteins were identified using PSORTb and TMHMM, followed by antigenicity and allergenicity predictions using VaxiJen and AlgPred. Linear B-cell epitopes and MHC-II binding sites were predicted using BepiPred and TepiTool, while physicochemical properties were analyzed with ExPASy ProtParam and Protein-Sol. The MEV construct was validated through molecular docking with TLR2 and TLR4 using HDOCK, revealing strong binding interactions. Molecular dynamic simulations confirmed the stability of the vaccine-receptor complexes, while PCA analysis indicated minimal conformational transitions. Immune simulations were conducted using C-ImmSim online software.

### Results

This study identified eight OMPs from *A. baumannii* strain VB7036 as potential immunogenic targets. MEV was designed using five critical B-cell epitopes from four proteins based on their antigenicity, non-allergenicity, and physicochemical properties. This MEV demonstrated strong binding to TLR2 and TLR4, indicating effective immune activation. Molecular dynamics simulations confirmed the structural stability of the MEV-TLR

**Data availability statement:** All supplementary data are available in the Supplemental File. The genome datasets analyzed during the current study are available in the GenBank database (https://www.ncbi.nlm.nih.gov/genbank/).

**Funding:** The author(s) received no specific funding for this work.

**Competing interests:** The authors have declared that no competing interests exist.

complexes. *In silico* immune simulations revealed that the MEV induced robust humoral and cell-mediated immune responses, including increased antibody production, T-cell activation, and cytokine release, suggesting the MEV's potential as an effective vaccine candidate for *A. baumannii*.

## Conclusion

This study developed an optimized MEV and identified novel drug targets against *A. baumannii*, providing broad protection against multidrug-resistant *A. baumannii* strains. MEV demonstrated significant potential due to its favorable physicochemical properties, as confirmed by molecular docking and dynamic simulations. However, more *in vitro* and *in vivo* studies are required to verify the drug's effectiveness.

## 1.  Introduction

*Acinetobacter baumannii*, a member of the ESKAPE (*Enterococcus faecium*, *Staphylococcus aureus*, *Klebsiella pneumoniae*, *A. baumannii*, *Pseudomonas aeruginosa*, and *Enterobacter* spp.) group of multidrug-resistant (MDR) organisms, poses a significant threat to healthcare settings by contributing to hospital-acquired infections and associated mortality [1,2]. This bacterium is known for its ability to develop resistance to multiple antibiotics, including aminoglycosides, fluoroquinolones, beta-lactams, and carbapenemals [3]. The prevalence of MDR strains of *A. baumannii* severely limits treatment options, making it a critical challenge for global health care. Despite the absence of a licensed vaccine against *A. baumannii*, various potential antigens have been identified through experimental animal studies, including outer membrane vesicles (OMVs), outer membrane protein A (OmpA), autotransporter (Ata), biofilm-associated protein (Bap), K1 capsular polysaccharide, and poly-N-acetyl-β-[1–6]-glucosamine (PNAG) [4]. However, the variability in these antigens and their absence in circulating strains limit their effectiveness against emerging *A. baumannii* strains, which may mutate under immunological pressure and downregulate the target antigens. Recent studies have used *in silico* prediction tools, reverse vaccinology, comparative genome analysis, and *in vitro* proteomics to identify potential vaccine targets [1,5]. Reverse vaccinology is a computational approach that analyzes bacterial genomes to identify and characterize promising antigens based on their antigenicity, allergenicity, and other relevant properties [6–8]. For example, *in silico* analysis has identified outer membrane β-barrel assembly protein (BamA), a highly conserved OMP involved in β-barrel assembly, as a promising vaccine candidate [5]. However, the emergence of new, extensively drug-resistant strains, possibly due to immune selection, leads to antigenic variability and reduced cross-protective efficacy [4]. The prevention and treatment of *A. baumannii* infections remain significant healthcare challenges because of the rapid emergence of MDR strains [9–12]. Due to the absence of effective antibiotics, vaccination has emerged as an optimal strategy for combating MDR pathogens.

Advances in computational biology and bioinformatics have enabled the rapid design of effective vaccine constructs, reducing reliance on traditional laboratory methods [13]. Using advanced computational tools and bioinformatics, we discover novel vaccine candidates, identify druggable targets, and propose strategies to combat *A. baumannii*-related infections. This approach highlights the expanding role of LBDD in addressing urgent global health challenges in infectious disease research. Peptide-based multi-epitope vaccines (MEVs) offer a breakthrough *in silico* methods to identify and target immunodominant epitopes, providing more efficient and targeted immune responses than traditional approaches [14]. MEVs,

which include multiple epitopes from different proteins, aim to elicit both humoral (antibody-mediated) and cellular (T cell-mediated) immune responses, offering broad protection against diverse *A. baumannii* strains [14]. This strategy reduces the likelihood of bacteria evading the immune response through mutations in a single antigen. Epitope-based immune-derived vaccines are generally safer than other vectors or attenuated live vaccines, providing essential T-cell help for antibody-directed vaccines and offering significant advantages over earlier vaccine designs [15]. Reverse vaccinology has effectively identified and designed vaccine targets against various pathogens, including *A. baumannii* [16]. Given these challenges, this study adopted a two-pronged approach: developing a novel MEV to elicit a robust immune response to combat *A. baumannii* infections. This study focused on essential proteins, virulence factors, and resistance determinants that contribute to the pathogenicity of this bacterium. Cytoplasmic proteins are typically targeted for small-molecule drug development, whereas membranes or OMPs are prioritized for vaccine development [17]. This strategy addresses the formidable pathogen's prevention and treatment options and offers valuable insight into designing and developing effective vaccines and therapeutics against *A. baumannii*.

## 2. Methods and materials

### 2.1. Consecutive analyses

**2.1.1. Sequence retrieval.** *A. baumannii* strain VB7036 was selected as a reference strain, previously isolated from a bacteremia patient's bloodstream in India in 2019 [18]. This strain is classified as sequence type 2

(ST2) under the Institut Pasteur multilocus sequence typing (MLST) scheme, a lineage known for its significant role in healthcare-associated infections. The selection of VB7036 is supported by its fully sequenced genome, which is publicly accessible under GenBank accession number CP050523, enabling a comprehensive analysis of its protein-coding sequences. Given its well-documented MDR profile and detailed genomic information, this strain is a robust model for identifying vaccine candidates against *A. baumannii*, particularly in addressing antimicrobial resistance within clinical settings.

**2.1.2 . Prediction of subcellular localization.** To enhance reproducibility, we used two computational tools with specified versions and settings to determine the subcellular localization of proteins from the *A. baumannii* strain VB7036. First, we used the PSORTb version 3.0.3 online server (https://www.psort.org/psortb/). The proposed tool is specifically designed to predict bacterial protein subcellular localization.

Additionally, the TMHMM Server version 2.0 (http://www.cbs.dtu.dk/services/TMHMM/) was used to predict transmembrane helices in proteins, which aids in distinguishing between cytoplasmic and membrane-associated proteins [19,20]. We configured the tool to evaluate Gram-negative bacteria, focusing on potentially extracellular outer membrane proteins (OMPs).

**2.1.3. Determination of antigenicity and allergenicity.** The antigenicity of probable immunogenic targets was predicted using the VaxiJen version 2.0 online server (http://www.ddg-pharmfac.net/vaxijen/VaxiJen/VaxiJen.html), with a classification threshold value set at > 0.5 [21]. Allergenicity was evaluated using the AlgPred version 2.0 web tool (https://webs.iiitd.edu.in/raghava/algpred2/batch.html), which uses a cutoff value of > 0.5 for the allergen prediction [22]. The tool was run using default configurations, and a hybrid approach that integrates various allergen prediction methods was applied.

**2.1.4. Assessing the similarity of immunogenic targets to the human proteome.** The homology of a subset of proteins from the human proteome (Humo sapiens, taxid:9606) was assessed using the BLASTp program available in the NCBI

database (https://blast.ncbi.nlm.nih.gov/). Proteins that exhibited similarities to the human proteome were removed from this investigation.

**2.1.5. Evaluating putative immunogenic target prevalence in diverse *A. baumannii* strains.** Initially, 560 *A. baumannii* genomes as a genome dataset were obtained from the GenBank database (https://www.ncbi.nlm.nih.gov/genbank/). The prevalence and conservation of proteins among these strains were assessed to induce a robust immune response to all *A. baumannii* strains. Homologs of each immunogenic target from the 560 *A. baumannii* strains were identified using BLASTp, and the sequences of each homolog were aligned using MegaX software [23]. Only those detected in over 75% of the strains were selected to ensure protein conservation. This approach enhances the probability of eliciting an immune response across various strains.

## 2.2. Immunoinformatics analyses

**2.2.1. Determination of linear B-cell epitopes and human MHC-II binding sites.** The BepiPred version 2.0 tool (http://www.cbs.dtu.dk/services/BepiPred/) was used to predict linear B cell epitopes within all identified potential immunogenic proteins. With a threshold score of > 0.6, the tool indicated potential epitopes based on the default epitope prediction algorithm [24]. The tool was run in its standard configuration, as recommended in the official documentation, and the ratio of B-cell epitopes to total amino acids was calculated for each protein.

TepiTool from the Immune Epitope Database (IEDB, http://tools.iedb.org/tepitool/help/) was employed [29] to predict human MHC-II binding sites. This tool was configured to use the top 5% of peptides predicted to bind to the 26 most prevalent MHC-II alleles, with all other settings remaining at their default values. The MHC-II binding sites were then normalized by calculating the ratio of binding sites to total amino acids for each protein [25].

**2.2.2. Physicochemical characteristics of putative immunogenic proteins.** Several computational tools were used for domain and protein characterization. The functional classes of the selected proteins were predicted using the VICMpred database v1.0 (http://webs.iiitd.edu.in/raghava/vicmpred/) [26] with default settings for the bacterial proteins. For domain identification, two resources were employed: the Conserved Domain Database (CDD) v3.19 in the NCBI Entrez system (https://www.ncbi.nlm.nih.gov/Structure/cdd/cdd.shtml), which annotates protein sequences with conserved domain positions [27], and the EggNOG v5.0 database (http://eggnog5.embl.de/), which provides orthology assignments and domain annotations across diverse species and viruses [28]. Additionally, the Expasy ProtParam server (https://www.expasy.org/protparam) (accessed using default parameters) was used to calculate the protein's amino acid composition, molecular weight (MW), and other physicochemical properties [29]. Detailed settings for each tool, including default configurations, were employed unless otherwise specified. These settings were selected to ensure comprehensive and consistent annotation across all analyzed protein sequences.

**2.2.3. Tertiary structure prediction and characterization of conformational B-Cell epitopes.** The 3D structures of the selected proteins were predicted using the Swiss Model server (https://swissmodel.expasy.org/), using homology modeling with default parameters unless specified otherwise [30]. To evaluate the quality of the predicted structures, the Protein Structure Analysis (ProSA) web server (https://prosa.services.came.sbg.ac.at/prosa.php) was employed to assess potential errors in the protein models based on their Z-scores. For further validation, SAVES v6.1 (https://saves.mbi.ucla.edu/) was used extensively, leveraging tools like ERRAT and Verify3D, with settings adjusted for the specific structural features of the proteins in question [31,32].

Conformational B-cell epitopes were predicted using the ElliPro server (http://tools.iedb.org/ellipro/) with a minimum score threshold of ≥ 0.8 and default settings for protrusion index analysis [33]. The predicted epitopes were visually represented using PyMOL version 2.3.4, where distinct colors were applied to highlight each epitope on the protein surface. PyMOL software was run, and image rendering was performed with high-quality settings to ensure clear visualization [34].

**2.2.4. Protein-Protein interaction using the STRING database.** We used STRING database v12.0 (https://string-db.org/) to assess hypothetical protein interactions with other *A. baumannii* proteins and infer their likely activities. Connection scores of at least 0.5 were considered significant. The analysis used default settings for evidence-based interaction sources, including experiments, databases, co-expression, neighborhood, gene fusion, and cooccurrence. The network edges were also set to confidence, and disconnected nodes were hidden to improve visualization.

## 2.3. Construction of multi-epitope vaccines

**2.3.1. Prediction and selection of optimal epitopes for vaccine development.** To identify potential targets for vaccine development, protein sequences were analyzed to identify linear B-cell epitopes using the Kolaskar and Tongaonkar method via the Antibody Epitope Prediction tool available on the IEDB analysis server v2.28 (http://tools.iedb.org). Default parameters were applied with the prediction threshold set to ≥ 0.35. The antigenicity of these potential immunogenic epitopes was assessed using the VaxiJen version 2.0 online server (http://www.ddg-pharmfac.net/vaxijen/VaxiJen/VaxiJen.html) with a threshold of ≥ 0.5 for predicted antigenicity. Allergenicity was evaluated using the AlgPred v2.0 server (https://webs.iiitd.edu.in/raghava/algpred2/batch.html), a hybrid approach and using a cut-off of ≥ 0.5 for allergen prediction. Additionally, the IEDB analysis server (http://tools.iedb.org/conservancy/) was used to assess epitope conservation and hydropathicity, where a 90% identity threshold was applied for conservation analysis, and the Kyte-Doolittle scale was used for hydropathicity scoring. Epitopes that demonstrated antigenicity lacked allergenicity, over 90% conservation, and had the lowest hydropathicity scores were chosen as promising candidates for subunit vaccine development.

**2.3.2. Structural construction and validation of the vaccine candidate.** To develop MEVs targeting *A. baumannii*, linear B-cell epitopes were connected using flexible GPGPG linkers. The arrangement with the highest antigenicity scores was determined using epitope shuffling methods. The 3D structures of the selected proteins were predicted using Swiss Model v2021-09-29 (https://swissmodel.expasy.org/) [30]. The following settings were employed: model building was performed using the default template search strategy, and homology models were constructed based on sequence identity. The quality and stability of these 3D models were then evaluated using Structure Validation S (SAVES) v6.1 (https://saves.mbi.ucla.edu/resultsjob=1243713&p=errat) [31,32], focusing on ERRAT for evaluating non-bonded interactions, VERIFY3D for assessing the structural environment, and PROCHECK for stereochemistry validation.

**2.3.3. Molecular docking.** In our docking study using HDOCK serve v2020.2 (http://hdock.phys.hust.edu.cn/) [35], the binding vaccine candidates for human Toll-like receptors 2 (TLR2, PDB: 2Z7X) and 4 (TLR4, PDB: 3FXI) were assessed. The docking process employed HDOCK's default rigid-body docking protocol, which uses a knowledge-based scoring function to rank docked conformations and refine them through iterative search optimization.

By default, this server is configured with a grid spacing of 1.200 Å to ensure precise sampling along the x, y, and z axes. To investigate different ligand orientations, rotational

sampling was performed using an Euler angle increment of 15.000°. The ligand's initial rotation was set to 0.000° on all axes, meaning no specific rotation was applied before docking. Docking scores were reported as binding energies in kcal/mol, where lower scores indicated stronger predicted binding affinities. Finally, the interactions within the docked complexes were visualized and validated using the PDBsum server v2021-12 (https://www.ebi.ac.uk/thornton-srv/databases/pdbsum/) [41], which provided detailed insights into hydrogen bonds, salt bridges, and hydrophobic contacts, further supporting the analysis of the molecular interactions.

**2.3.4. Evaluation of the allergenicity, antigenicity, solubility, and physicochemical properties of a multi-epitope vaccine.** To evaluate the allergenicity of the vaccine constructs, we used the AlgPred v2.0 server (https://webs.iiitd.edu.in/raghava/algpred2/batch.html) with a cutoff value of 0.5 or higher for allergenicity predictions. AlgPred v2.0, a web-based tool, uses a hybrid approach that integrates machine learning methods with allergen databases to predict allergenicity. Additionally, allergenicity analysis was performed using AllerTOP v2.0 (https://www.ddg-pharmfac.net/AllerTOP/), in which the prediction model employs auto-cross-covariance transformation of protein sequences into vectors. Both tools consistently confirmed that the vaccine construct was nonallergenic, minimizing the risk of allergic reactions during vaccination protocols [22,36].

For antigenicity prediction, we employed the VaxiJen v2.0 server (http://www.ddg-pharmfac.net/vaxijen/VaxiJen/VaxiJen.html), which uses a target threshold of 0.4 to indicate probable antigenicity [37].

Next, we used the Protein-Sol v2.0 server (https://protein-sol.manchester.ac.uk/) to predict the solubility of engineered MEV, applying a cutoff score of 0.45 to indicate solubility. The server estimates solubility based on experimental datasets and sequence information, providing results for each amino acid residue along the protein sequence [38].

For further characterization of the MEV, the ExPASy ProtParam tool (https://web.expasy.org/protparam/) was used to predict fundamental physicochemical properties, including MW, theoretical isoelectric point (pI), protein half-life, aliphatic index, instability index, and GRAVY score. The tool's default parameters were used for sequence input, and the results were cross-checked for accuracy [39,40]. These parameters are critical for understanding the structural and functional properties of MEVs.

**2.3.5. Molecular dynamics (MD) simulation of the selected Multi-Epitope Vaccine in complex with immune receptors.** MD simulations were performed to assess the feasibility of interaction between the chimeric MEV and TLR2 and TLR4 complexes using GROMACS 2019 software [41]. The simulations employed the Optimized Potential for Liquid Simulations force field to evaluate the stability and conformational dynamics of MEV both in its unbound form and in complex with its receptors. Each complex was placed in a 10 Å solvent box filled with simple point-charge water molecules, and system neutrality was achieved by adding appropriate amounts of $Na^+$ and $Cl^-$ ions. Energy minimization was followed by two-phase equilibration: the systems were equilibrated for 100 ps under a constant number of particles, volume, and temperature and a continuous number of particles, pressure, and temperature. The Parrinello–Rahman barostat was used to maintain a stable temperature of 300 K and pressure of 1.0 bar. Long-range electrostatic interactions were handled by the particle mesh Ewald method with a 10 Å cutoff and grid spacing of 0.16 nm.

In contrast, van der Waals interactions were calculated using a 1 nm cutoff. The Linear Constraint Solver (LINCS) algorithm was applied to constrain covalent bond lengths.

MD simulations are instrumental in elucidating the stability and dynamics of vaccine complexes. This study investigates the stability of these complexes over a simulation period of 100 ns, focusing on key metrics including root mean square fluctuation (RMSF), root mean

square deviation (RMSD), radius of gyration (Rg), and the number of hydrogen bonds within each protein-protein complex was also evaluated to assess the stability of interactions within the vaccine complex [42].

**2.3.6. Principal Component Analysis (PCA) of conformational transitions in vaccine candidate-TLR2 and TLR4 complexes.** PCA was performed to investigate the conformational space and transitions in the vaccine candidate and its complexes with TLR2 and TLR4. PCA calculates the covariance matrix of the positional fluctuations of the backbone atoms, which provides insights into the dominant motion patterns of the system. The vaccine candidate's first principal components (PC1 and PC2), vaccine-TLR2 and vaccine-TLR4 complexes, were obtained by projecting the trajectories onto their respective eigenvectors. This projection visualizes different conformations in a two-dimensional (2D) space. The PCA analysis was conducted on molecular dynamics (MD) simulations throughout 100 ns, examining the conformational transitions and stability of the complexes.

**2.3.7. Immune simulation.** The C-ImmSim server (https://kraken.iac.rm.cnr.it/C-IMMSIM/index.php) was used to characterize the profile of the stimulated immune response by the MEV [32]. An injection containing 1000 vaccine molecules was administered to stimulate an immune response in humans. The simulation parameters were set with a random seed of 12,345, a simulation volume of 10 μl, and a simulation step of 100.

## 3. Results

### 3.1. Selection of the surface-exposed, antigenic and non-allergen proteins

The proteome of the *A. baumannii* strain VB7036 comprises 3752 proteins. Subcellular localization analysis identified 138 proteins on the cell surface. A comprehensive characterization of the 138 extracellular and OMPs of the *A. baumannii* strain VB7036 is provided in the S1 Table.

Of these, 68 proteins exhibited antigenicity with a score ≥ 0.5. Further analysis using Alg-Pred identified 57 non-allergic proteins. The study workflow has been illustrated in Fig 1.

### 3.2. Selection results of non-homologous proteins to the human proteome for vaccine targeting

All proteins mentioned above were evaluated to assess their similarity to Homo sapiens (taxid:9606). As a result, ten homologous proteins were excluded, leaving 47 non-homologous human proteins.

### 3.3. Prevalence of putative immunogenic targets in circulating a. baumannii strains

The prevalence of 47 selected proteins was checked across 560 *A. baumannii* strains. Three proteins with low prevalence ( < 75%) were excluded from this study.

### 3.4. Selection of proteins with high ratio of MHC-II binding sites

Our analysis indicated that out of the 44 proteins, 13 exhibited a ratio of linear B-cell epitopes to human MHC-II-binding sites of less than 0.5 and MWs of less than 110 kDa. Additionally, none of the proteins had MWs exceeding 110 kDa.

### 3.5. Physicochemical properties of potentially immunogenic proteins

Five of the 13 proteins analyzed were excluded because of their physicochemical characteristics. Eight proteins were identified as potential vaccine candidates (Fig 2). Table 1 presents the

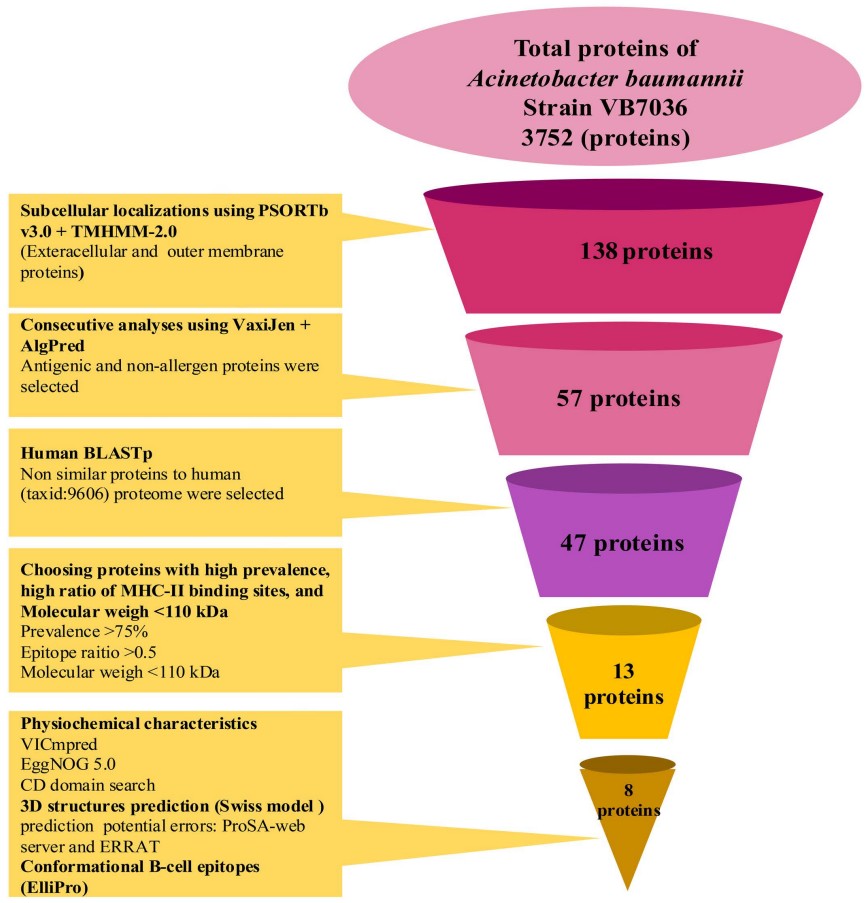

**Fig 1. (A) Workflow for identifying novel putative immunogenic targets against *Acinetobacter baumannii* using a reverse vaccinology approach.** The initial dataset comprised 3752 proteins from the *A. baumannii* strain VB7036 Proteome. Subcellular localization was predicted using PSORTb v3.0 and TMHMM-2.0 to identify extracellular and outer membrane proteins (OMPs). Next, VaxiJen and AlgPred are applied to select antigenic and non-allergenic proteins. BLASTp analysis is then used to remove proteins similar to the human proteome (taxid: 9606). Further refinement includes selecting proteins with a high prevalence of MHC-II binding sites, a high epitope ratio, and a molecular weight of 110 kDa. The final selection process involved physicochemical characterization, 3D structure prediction using the Swiss Model, and conformational B-cell epitope prediction using ElliPro.

physiological properties of eight putative immunogenic proteins from *A. baumannii*. Table 2 presents the detailed results of the conserved domains of these proteins identified using the CDD and EggNOG.

## 3.6. Tertiary Structures Of Vaccine Candidate Proteins And Their Conformational Epitopes

SWISS-MODEL and PorSA analyses indicated acceptable folding for all eight proteins (S1 Fig). The tertiary structures of these proteins were predicted, and their conformational epitopes were illustrated based on their tertiary structures. The number of conformational B-cell epitopes identified for each protein was as follows: MlaD superfamily (WP_000842362.1): three epitopes; multidrug efflux resistance-nodulation-division (RND) transporter outer membrane channel subunit AdeK (WP_001174793.1): five epitopes, OmpA family protein (WP_000836015.1): three epitopes; nlpD superfamily (WP_000550750.1): five epitopes,

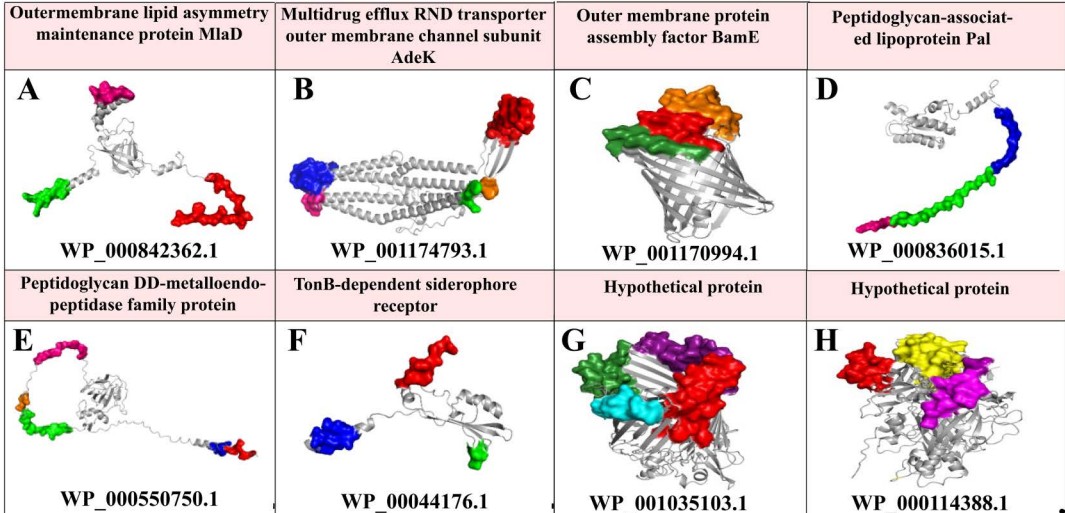

**Fig 2. Conformational B-cell epitopes were identified and mapped onto the tertiary structure of selected *Acinetobacter baumannii* proteins using ElliPro and visualized using PyMOL software.** Each distinct color represents a conformational B cell epitope predicted on the protein surface, indicating potential immunogenic regions. The proteins included the following: **(A)** Outer membrane lipid asymmetry maintenance protein MlaD (WP_000842362.1), **(B)** Multidrug efflux RND transporter, outer membrane channel subunit AdeK (WP_001174793.1), **(C)** Outer membrane protein assembly factor BamE (WP_001170994.1), **(D)** peptidoglycan-associated lipoprotein Pal (WP_000836015.1), **(E)** peptidoglycan DD-metallo-endopeptidase family (WP_000550750.1), **(F)** TonB-dependent siderophore receptor (WP_00044176.1), **(G)** Hypothetical protein 1 (WP_001035103.1), and **(H)** Hypothetical protein 2 (WP_000114388.1).

PRK13524 superfamily protein (WP_00044176.1), four epitopes, hypothetical protein (WP_001035103.1): three epitopes, hypothetical protein (WP_000114388.1): three epitopes, and OMP assembly factor BamE (WP_001170994.1): three epitopes (Fig 2).

## 3.7. Analysis of Protein-Protein Interaction Networks for Identifying Putative Immunogenic Candidates

Based on the CDD and EggNOG analyses, the functions of two vaccine candidate proteins (WP_001035103.1 and WP_000114388.1) were undetected and remained unidentified. STRING analysis revealed that WP_001035103.1 contained several proteins-neighborhood interactions, including three hypothetical proteins (AGQ13058, AGQ13053.1, and AGQ13056.1), a long-chain fatty acid transport protein (AGQ13057.1), two putative double-glycine peptidases (AGQ13054.1 and AGQ13055.1), and an anthranilate/para-aminobenzoate synthase component (AGQ13052.1). According to the STRING database, WP_000114388.1 exhibited neighborhood interactions with two hypothetical proteins (AGQ13058.1 and AGQ13059.1), a long-chain fatty acid transport protein (AGQ13057.1), and putative double-glycine peptidases (AGQ13054.1 and AGQ13055.1). It also exhibited interactions with four hypothetical proteins (AGQ12916.1, AGQ14260.1, AGQ14262.1, and AGQ14259.1) and a lipase chaperone (LifO) (Fig 3).

## 3.8. Analysis and Evaluation of Multi-Epitope Vaccine Construct Development

Of the eight proteins studied, five B-cell epitopes from four proteins showed potential for inclusion in the final vaccine design (Table 3) based on stringent criteria, including high

**Table 1. Physicochemical properties of eight putative immunogenic proteins of A. baumannii.**

| Accession number | Numbers of amino acids | Molecular weight (kDa) | Theoretical PI | Subcellular localization | Functional class | TMH | Estimated half-life (E. coli) | Stability | Instability index | Aliphatic index | Hydropathicity | Allergenicity score | Similarity to the human proteome | Antigenicity score | No. of linear B-cell epitopes | B cell epitope ratio |
|---|---|---|---|---|---|---|---|---|---|---|---|---|---|---|---|---|
| WP_000842362.1 | 226 | 24.11 | 4.98 | Outer membrane | Cellular process | 1 | >10 hours | Stable | 20.14 | 88.05 | -0.085 | -0.23 | No | 0.7338 | 6 | 0.63 |
| WP_001174793.1 | 484 | 52.8 | 9.02 | Outer membrane | Cellular process | 0 | >10 hours | Stable | 39.59 | 92.02 | -0.311 | -0.21 | No | 0.6450 | 20 | 0.73 |
| WP_000836015.1 | 187 | 20.07 | 6.91 | Outer membrane | Metabolism Molecule | 0 | >10 hours | Stable | 32.96 | 74.76 | -0.443 | 0 -0.35 | No | 0.8113 | 8 | 0.56 |
| WP_000550750.1 | 276 | 29.62 | 9.84 | Outer membrane | Metabolism Molecule | 0 | >10 hours | Stable | 37.24 | 82.39 | -0.205 | -0.27 | No | 0.6465 | 6 | 0.57 |
| WP_00044176.1 | 754 | 82.82 | 5.66 | Outer membrane | Cellular process | 0 | >10 hours | Stable | 35.31 | 75.38 | -0.573 | 00.24 | No | 0.8130 | 28 | 0.6 |
| WP_001035103.1 | 649 | 70.86 | 6.27 | Extracellular | Virulence factors | 0 | >10 hours | Stable | 26.41 | 84.76 | -0.285 | 0.27 | No | 0.6426 | 24 | 0.56 |
| WP_000114388.1 | 386 | 41.61 | 4.45 | Outer membrane | Cellular process | 0 | >10 hours | Stable | 39.73 | 74.56 | -0.334 | 0.27 | No | 0.8203 | 15 | 0.58 |
| WP_001170994.1 | 132 | 14.36 | 9.04 | Outer membrane | Cellular process | 0 | >10 hours | Stable | 26.63 | 95.30 | 0.002 | 0.23 | No | 0.4154 | 15 | 0.46 |

**Table 2. Conservation of the conserved domains of putative immunogenic proteins against *A. baumannii* using the NCBI Conserved Domain Database and EggNOG.**

| Accession number | Extracted Names from NCBI | EGGNOG5 | CD-search |
|---|---|---|---|
| WP_000842362.1 | Outer membrane lipid asymmetry-maintenance protein MlaD | Secondary metabolite biosynthesis, transport, and catabolism (ABC-type transport system involved in resistance to organic solvents periplasmic component) | MlaD superfamily |
| WP_001174793.1 | Multidrug efflux RND transporter outer membrane channel subunit AdeK | Cell wall/membrane/envelope biogenesis (RND efflux system, outer membrane lipoprotein) | Efflux_OM_AdeK superfamily |
| WP_000836015.1 | Peptidoglycan-associated lipoprotein | Cell wall/membrane/envelope biogenesis It belongs to the OmpA family | OmpA family proteins |
| WP_000550750.1 | Peptidoglycan DD-metalloendopeptidase family | Cell wall/membrane/envelope biogenesis (Peptidase family M23) | NlpD superfamily |
| WP_00044176.1 | TonB-dependent siderophore | Inorganic ion transport and metabolism (TonB-dependent Receptor Plug) | PRK13524 superfamily |
| WP_001035103.1 | Hypothetical protein | Not Available | Not Available |
| WP_000114388.1 | Hypothetical protein | Not Available | Not Available |
| WP_001170994.1 | Outer membrane protein assembly factor BamE | Cell wall/membrane/envelope biogenesis (Part of the outer membrane protein assembly complex, which is involved in the assembly and insertion of beta-barrel proteins into the outer membrane) | BamE |

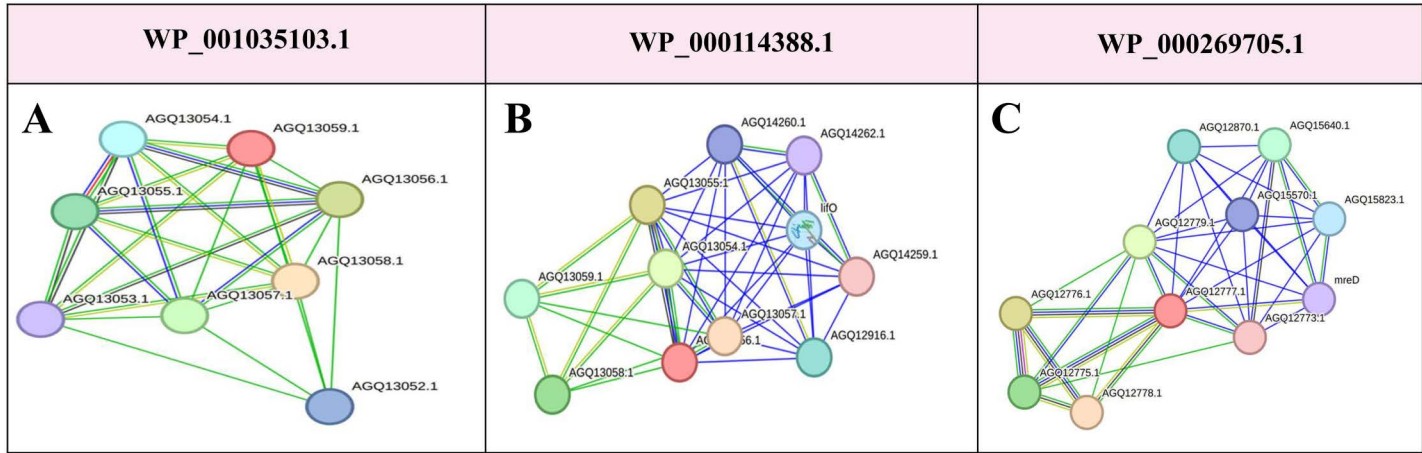

**Fig 3. Protein-protein interactions (PPIs) for the following proteins were analyzed using the STRING tool:** (A) WP_001035103.1: Hypothetical protein with predicted interactions within a complex network, indicating potential involvement in essential biological processes. (B) WP_000114388.1: Hypothetical protein interacting with proteins involved in critical cellular functions. (C) WP_000269705.1 is an unknown protein with multiple interactions, suggesting cellular structural integrity and involvement in essential pathways. Each node in the interaction network represents a protein, and the edges (lines) between nodes indicate predicted or known interactions. The line thickness reflects the strength of the interactions, and different colored lines represent distinct interaction types (e.g., direct physical binding or functional association). The figure highlights the complexity of protein interactions for these hypothetical proteins, providing insights into their potential roles in *A. baumannii* biology.

antigenicity (VaxiJen scores ≥ 0.5), non-allergenicity (confirmed by AllerTOP and AlgPred), 100% sequence conservancy across multiple *A. baumannii* strains, optimal epitope lengths (8–20 amino acids), strong binding affinity to MHC molecules (IC50 values < 100 nM), appropriate molecular weight, minimal toxicity, no cross-reactivity with human proteins, favorable physicochemical properties (solubility score: 0.844), minimal transmembrane regions, structural integrity, and ease of synthesis and expression. The arrangement with the highest

**Table 3. Five promising epitopes from four selected *A. baumannii* OMPs were used in the MEV.**

| Accesion number | Start | End | Epitope | Antigenicity (Score) | Allergenicity | Conservancy |
|---|---|---|---|---|---|---|
| WP_001174793.1 | 133 | 140 | LATQSARD | Ag (0.6102) | Non-allergen | 100% |
| | 322 | 347 | LPIFDWGTRRANVKISETDQKIALSD | Ag (0.9824) | Non-allergen | 100% |
| WP_001035103.1 | 5 | 39 | SSTINEDPNSGTNNGNLTSGSCTPTTSDNGAEDST | Ag (1.5996) | Non-allergen | 100% |
| WP_000114388.1 | 204 | 222 | KTGDSPYEIGLDELSTGKG | Ag (1.0887) | Non-allergen | 100% |
| WP_00044176.1 | 102 | 127 | NSRNSVRYGWKGERDTRGDSNWVPAE | Ag (1.3596) | Non-allergen | 100% |

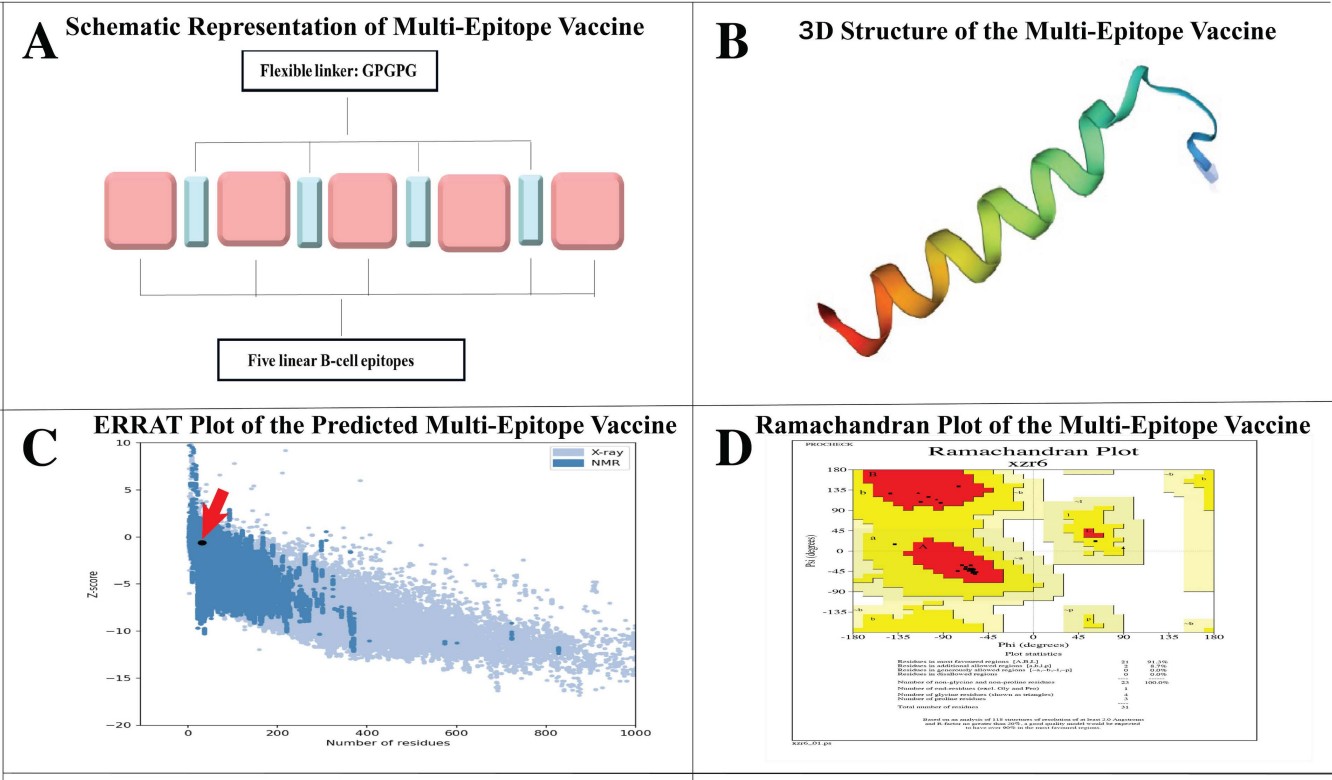

**Fig 4. Prediction and Validation of 3D Structures of MEVs. (A)** Schematic representation of multi-epitope vaccine components, illustrating the arrangement of five linear B-cell epitopes joined together using GPGPG flexible linkers. This arrangement enhanced the immunogenic response by improving epitope presentation and flexibility. **(B)** The MEV's Tertiary structure was predicted using the Swiss Model. It shows the linked epitopes' overall folding and structural conformation. The distinct color highlights different protein regions to visualize how epitopes are arranged in 3D space. **(C)** ERRAT plot of the overall quality of the predicted MEV structure. The ERRAT score provides insights into the reliability of structures based on non-bonded atomic interactions, indicating regions with higher structural accuracy. **(D)** Ramachandran plots of the MEVs were used to validate the stereochemical quality of the predicted structure. The plot assesses the phi (φ) and psi (ψ) dihedral angles of the residues, with most residues falling within the favorable regions, confirming the structural stability of the MEV.

antigenicity scores was determined using epitope shuffling methods (S2 Table). These epitopes were linked using a GPGPG linker (Fig 4A). The vaccine sequence was converted into the FASTA format and evaluated based on several criteria, such as antigenicity, non-allergenicity, and solubility. A schematic representation of the final MEV peptide is presented in Fig 4B. Using the SAVES server, the vaccine achieved an ERRAT of 85%. ProSA-web analysis of the MEV constructs is shown in Fig 4C. Fig 4D shows 91.3% Ramachandran-favored residues and 8.7% in additional permissible regions for MEV. The VaxiJen server predicted an antigenicity

score of 1.4514 for MEV, confirming it as nonallergenic using AlgPred 2.0 and AllerTOP v2.0. The selected vaccine candidate exhibited the highest solubility score (0.844). MEV had a low molecular weight (13.69 kDa) and high thermotolerance, as indicated by an aliphatic index of 42.99. Its pI was 4.53 and was identified as hydrophilic owing to its negative GRAVY score of -1.036. The instability index of the vaccine was 27.28, indicating that it is a stable polypeptide. MEV's estimated half-life is 1.9 hours *in vitro* in mammalian reticulocytes and > 10 hours *in vivo* in *E. coli*. MEV interacted with both TLR2 and TLR4, showing favorable binding affinities, as indicated by the docking scores.

## 3.9. Molecular docking

We used HDOCK for docking analysis and ranked the top 10 models according to docking score, confidence score, ligand RMSD (Å), and interface residues. Models with highly negative docking scores, higher confidence scores, and lower RMSD values were prioritized to ensure binding solid affinity and prediction reliability. The top-ranked models demonstrated strong docking scores, high confidence, and favorable RMSD values. The top model for the MEV-TLR2 interaction had a docking score of -232.85 kcal/mol, whereas the MEV-TLR4 interaction score was -243.55 kcal/mol, indicating a stronger affinity for TLR4. The confidence scores were 0.8398 and 0.8666 for TLR2 and 0.8666 for TLR4, with higher confidence in TLR4 interactions. The ligand RMSD values were 91.56 Å for TLR2 and 48.85 Å for TLR4, indicating better docking precision for TLR4. The TLR-2/MD complex shows three hydrogen bonds, 262 non-bonded, and one salt bridge were observed. The protein-protein interaction of construct and TLR2/MD showed that IIe111-Arg155 amino acids have two H-bond and Trp114-Thr84 have one H-bond, make contact along with other interactions, as highlighted in Fig 5A. The

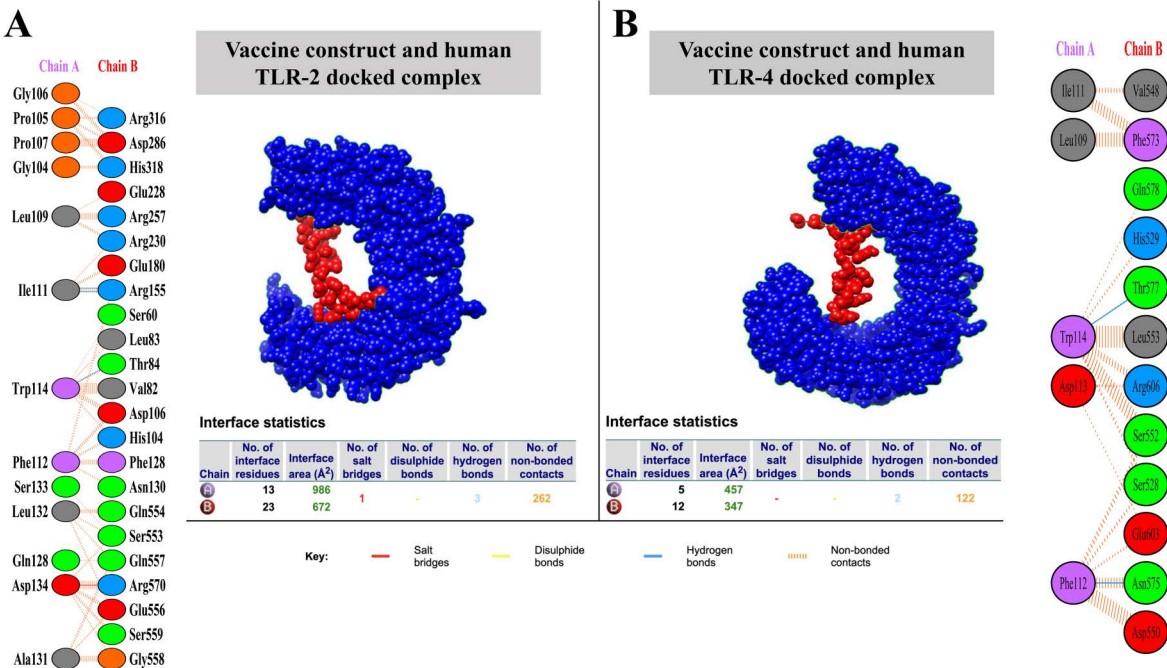

**Fig 5. Interaction details between the MEV and Toll-Like Receptors (TLRs).** (**A**) Interaction between the MEV candidate and TLR-2 (Vaccine construct and human TLR-2 docked complex). (**B**) Interaction between the MEV candidate and TLR-4 (Vaccine construct and human TLR-4 docked complex). Non-bonded interactions between the vaccine model and TLR2-4/MD protein are shown in orange. The blue line represents the hydrogen bond between the docked complex.

TLR-4/MD complex shows two hydrogen bonds and 122 non-bonded interactions, while no salt bridge was observed. The protein-protein interaction of construct and TLR2/MD showed that Trp114-Thr577 amino acids have one H-bond to interact with other interactions, as highlighted in Fig 5B.

These interactions indicate that the MEV can effectively activate innate immune responses, which is crucial for initiating downstream adaptive immunity.

### 3.10. Molecular dynamics simulation of the MEV, TLR2-MEV, and TLR4-MEV complexes

The global structural stabilities of the MEV, TLR2-MEV, and TLR4-MEV systems were evaluated using several MD simulation parameters. The RMSD was used to measure the conformational changes in each system. The RMSD values for the TLR2-MEV and TLR4-MEV complexes remained stable throughout the simulation, ranging between 0.4 and 0.6 nm, indicating minimal structural deviations and suggesting that both systems achieved stability early in the simulation and maintained stability after that (Fig 6A). RMSF analysis further confirmed the stability of the TLR2-MEV and TLR4-MEV complexes, with no significant fluctuations observed in individual residues across the protein structures (Fig 6B). Finally, the compactness of the structures was evaluated using Rg, which remained stable at approximately 2.5 nm for both the TLR2-MEV and TLR4-MEV systems, indicating consistent

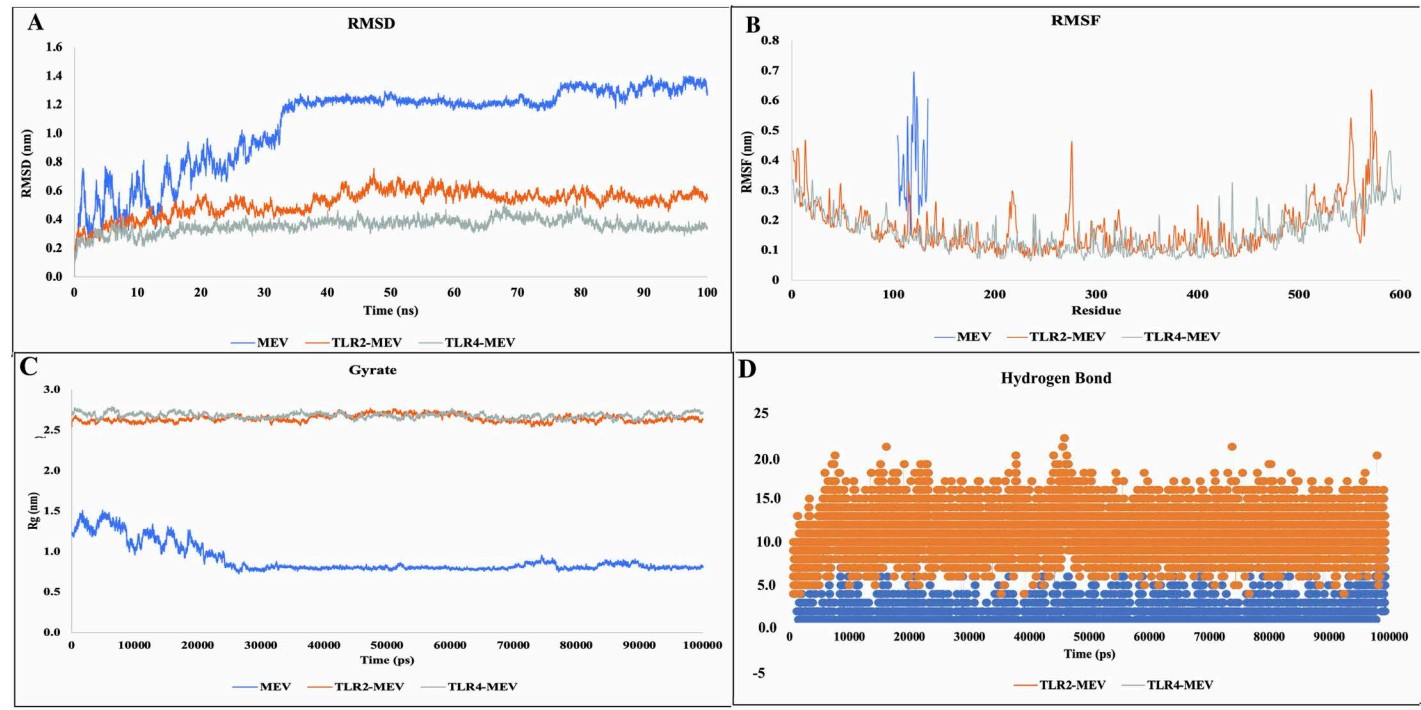

**Fig 6. Structural Stability and Flexibility Analysis of MEV, TLR2-MEV, and TLR4-MEV Systems During Molecular Dynamics Simulation. (A)** Root Mean Square Deviation (RMSD) plots of the MEV, TLR2-MEV, and TLR4-MEV systems over 100 ns of molecular dynamics simulation, indicating the overall structural stability of the complexes. TLR2-MEV and TLR4-MEV exhibit stable RMSD values in the 0.4–0.6 nm range, whereas MEV alone exhibits higher deviations. **(B)** Root mean square fluctuation (RMSF) plot showing residue flexibility for the three systems. The TLR2-MEV and TLR4-MEV complexes exhibit minimal fluctuations across the residues, whereas the MEV system exhibits increased flexibility around residues 100-150. **(C)** The radius of gyration (Rg) plot depicts the compactness of the systems over time. The TLR2-MEV and TLR4-MEV complexes remained compact with an Rg of approximately 2.5 nm, whereas the MEV system showed a decrease in Rg, stabilizing near 1.0 nm. These results indicate the enhanced structural stability of the MEV complexes in the presence of TLR2 and TLR4. **(D)** Number of hydrogen bonds formed between TLR2 and TLR4 and vaccine construct during MD simulation.

structural compactness throughout the simulation (Fig 6C). These results demonstrate that the interactions of MEV with TLR2 and TLR4 promote and maintain the structural stability of the complexes during MD simulations.

The stability of the vaccine molecules was further assessed by analyzing the total number of hydrogen bonds formed throughout the simulation period. The formation and fluctuations of hydrogen bonds, along with any significant changes in the complexes, were used to evaluate the rigidity of the interactions. Both the vaccine-TLR2 and vaccine-TLR4 complexes exhibited stable hydrogen bond numbers, as shown in Fig 6D, suggesting the relative stability of the complexes.

### 3.11. PCA analysis of conformational transitions in vaccine-TLR complexes

The Principal Component Analysis (PCA) of the 100 ns molecular dynamics (MD) trajectory revealed that the vaccine candidate and its complexes with TLR2 and TLR4 exhibit stable conformational behavior, with all three systems occupying a small, well-defined region in phase space. As shown in Fig 7 (Fig 7A, Fig 7B for the vaccine-TLR2 complex, and Fig 7C for the vaccine-TLR4 complex), the 2D projections of the trajectories demonstrate minimal fluctuations, indicating that the structures remain stable throughout the simulation. The trajectory points for each system are tightly clustered, suggesting limited conformational transitions. Furthermore, the analysis revealed a balance between correlated and anti-correlated motions within the complexes, indicative of a stable conformational space. These results, supported by other MD simulation parameters, confirm the structural stability of the vaccine candidate and its interactions with TLR2 and TLR4.

### 3.12. Immune simulation by the vaccine structure

The *in silico* simulation results revealed that the designed vaccine effectively induces both humoral and cell-mediated immune responses. Increased antibody levels and B cell populations marked the humoral immune response. B cells displayed a strong early activation, peaking around days 10–15, followed by a transition to memory and presenting states. Active B cells gradually declined as memory B cells became predominant. Plasmablasts peaked around day 10, initially producing IgM, which shifted to IgG1 and IgG2 isotypes over time. This progression was reflected in antibody production, where antigen levels peaked early, followed by a sustained increase in IgM and IgG antibodies. IgG maintained higher levels over time, indicating the development of long-term humoral immunity.

T cell populations, including cytotoxic T cells and helper T cells (Th cells), showed increased responses aligned with the formation of memory cells. Early in the response, active T cells surged, then gradually declined as memory T cells stabilized. Cytotoxic T cells and helper T cells exhibited distinct patterns, with active states dominating initially and memory subsets increasing over time. Helper T cell subsets, such as Th1, Th2, and regulatory T cells (Tregs), demonstrated unique activation profiles. Tregs steadily increased, suggesting their role in maintaining immune homeostasis, while Th1 and Th2 subsets showed specific expansion patterns tailored to the immune challenge.

The activity of macrophages, dendritic cells, and natural killer (NK) cells remained consistent throughout the 35 days. Additionally, significant increases in IFN-γ, IL-2, and TGF-β levels were observed following subsequent exposure, indicating that the vaccine triggered robust immune responses. Cytokine dynamics showed temporally distinct peaks, highlighting their roles in regulating the immune response. IL-2 peaked early, corresponding to the activation of T cells, while IFN-γ and TNF-α exhibited prolonged activity, supporting sustained immune

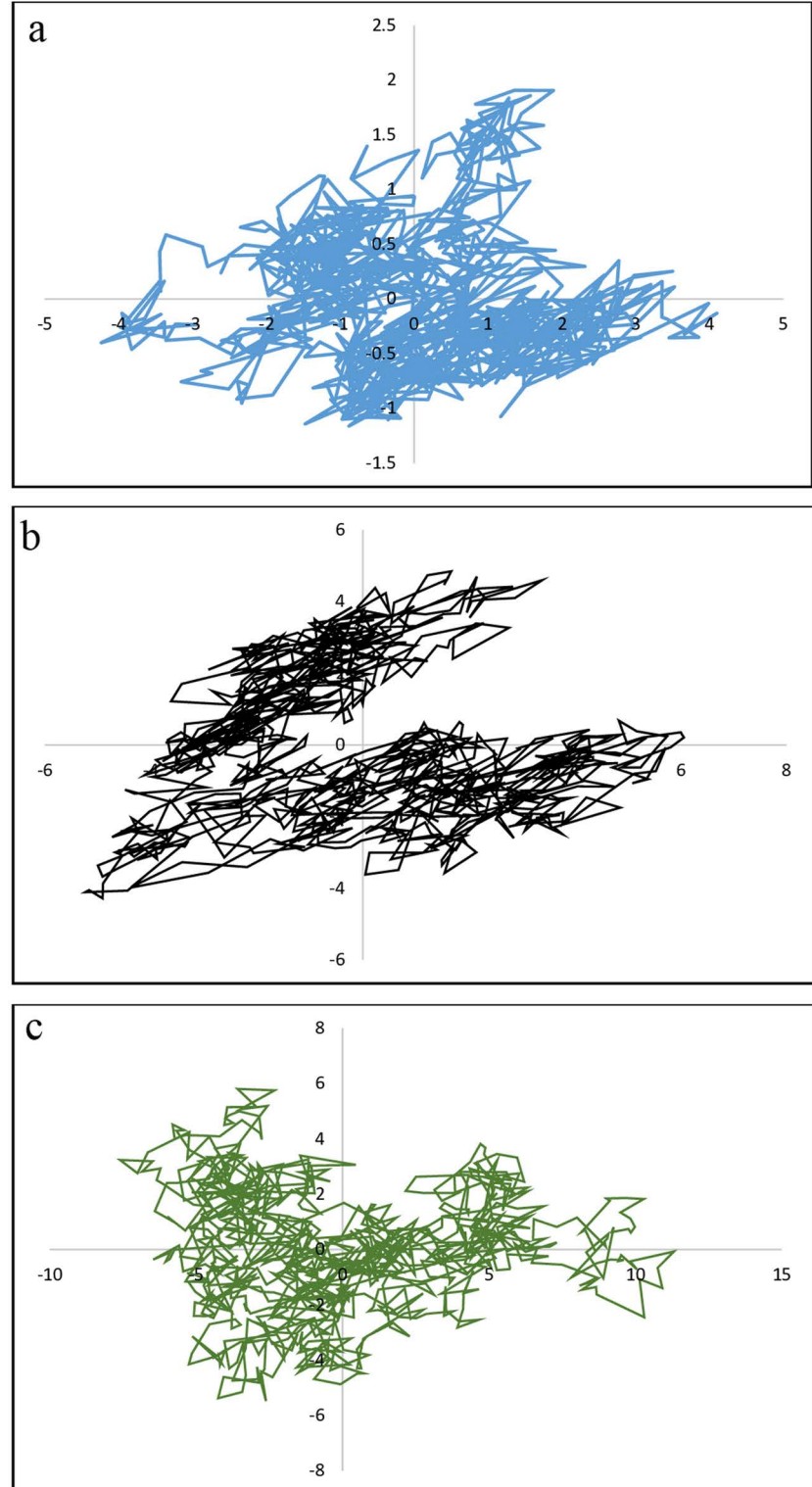

**Fig 7.  Projection of motion in phase space through Principal Component Analysis (PCA) of the 100 ns MD trajectory.** The 2D projections represent the conformational sampling of the vaccine candidate (**A**), vaccine-TLR2 complex (**B**), and vaccine-TLR4 complex (C) along the first two principal components (PC1 and PC2). Each panel displays the system's trajectory throughout the simulation, with the points clustered within a small, defined region of phase space, indicating stable conformations. The compact clustering of points in all three systems suggests minimal

fluctuations and limited conformational transitions. The analysis indicates balanced space occupation with low volatility, further supporting the stability of the structures and confirming that the vaccine candidate and its complexes with TLR2 and TLR4 remain structurally stable throughout the simulation period.

functions. IL-4 and IL-6 had delayed but extended responses, promoting antibody class switching and B cell differentiation. TNF-α dominated during the early inflammatory phase, while IFN-γ was critical in sustaining adaptive immunity. These cytokine patterns underscored the highly regulated immune signaling necessary for effective and balanced immune responses (Fig.8).

## 4. Discussion

*A. baumannii* has emerged as a formidable pathogen because of its rapid acquisition of broad-spectrum antimicrobial resistance genes, which result in MDR strains [43]. Resistance to antibiotics reduces the effectiveness of current antibiotic treatments, highlighting the urgent need for novel therapeutic strategies, including vaccines, to control and prevent *A. baumannii* infections.

Given the lack of licensed vaccines against *A. baumannii*, vaccine development remains a priority. Advances in vaccine research have focused on MEVs using recombinant proteins or peptide fragments to target essential pathogenic components, resulting in safer and more

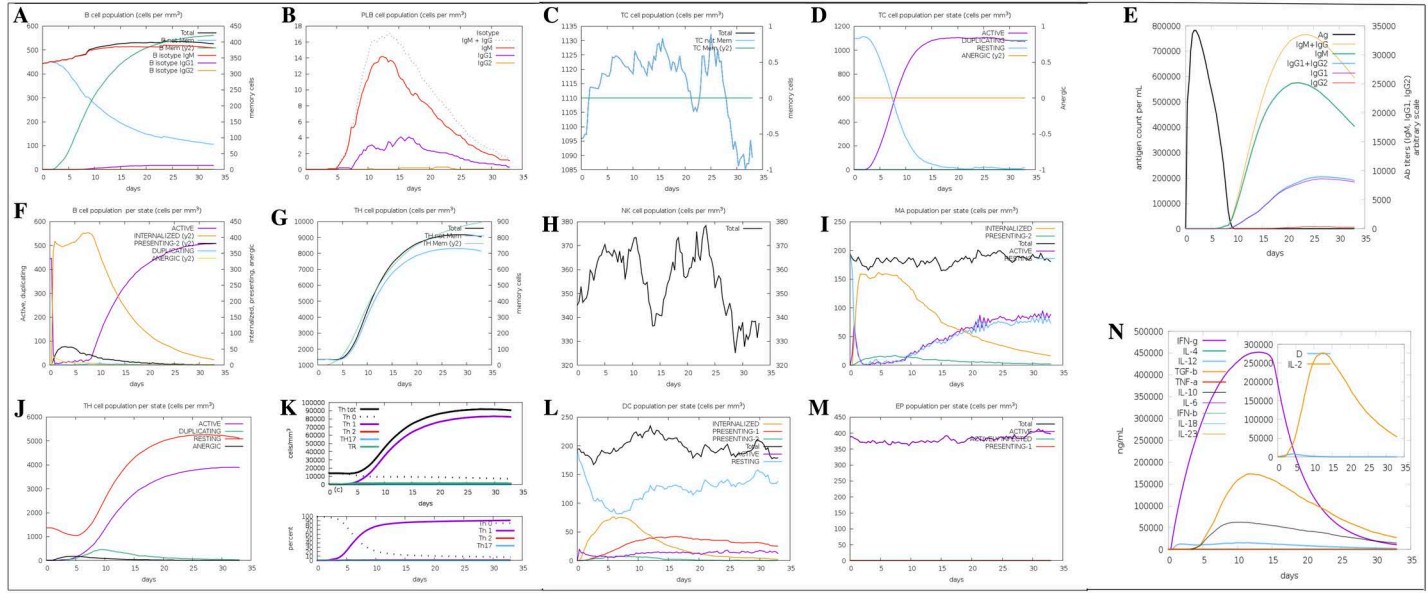

**Fig 8. Simulation of the immune responses to vaccination by MEV. (A)** B-cell population dynamics: Total B-cell population and subpopulations (IgM, IgG1, IgG2) showing early IgM dominance and later IgG increase. **(B)** Plasma B-cell dynamics: B-cell subsets produce IgM, IgG1, and IgG2 peaks during different response phases. **(C)** Cytotoxic T-cell dynamics: Total Tc cell population and memory Tc cells fluctuate, peaking during the response phase. **(D)** TC cell states: TC cells transition through active, duplicating, resting, and anergic states over time. **(E)** Antibody dynamics: Antigen clearance correlates with early IgM and later IgG1/IgG2 antibody production. **(F)** B-cell states: B-cells transition from presenting to active and anergic states during the immune response. **(G)** TH cell dynamics: Total and memory TH cell populations steadily increase, reflecting long-term immunity. **(H)** NK cell dynamics: NK cell populations fluctuate, reflecting their role in early immune response modulation. **(I)** Macrophage states: Active macrophages dominate during the early phase, with presenting and resting states stabilizing later. **(J)** TH1 cell states: Active TH1 cells peak early, supporting a Th1-driven immune response. **(K)** T-helper cell subtypes: Th1 cells dominate, with contributions from Th2, Th17, and Treg populations. **(L)** Dendritic cell states: DCs in presenting and internalized states dominate during antigen presentation. **(M)** Effector T-cell states: Active TE cells dominate during the response, with duplicating and anergic states appearing later. **(N)** Cytokine dynamics: IFN-γ peaks early, signaling a Th1-driven response, with IL-2, TNF-α, and other cytokines supporting inflammation and adaptive immunity.

effective solutions. This study aimed to develop an MEV targeting OMPs critical for bacterial interactions, pathogenicity, and immune evasion. OMPs are ideal vaccine targets because of their accessibility and vital role in bacterial function.

We employed *in silico* methods to systematically assess *A. baumannii* OMPs and identify surface-exposed, conserved B cell epitopes that likely elicit robust immune responses.

Our study has identified several immunogenic proteins and epitopes as potential vaccine candidates for *A. baumannii*. OMPs, such as TonB-dependent receptors and RND efflux transporters like AdeK, have emerged as promising targets. TonB-dependent receptors play a crucial role in iron uptake and bacterial virulence, while AdeK is involved in efflux pump activity, which contributes to MDR. Targeting these proteins could disrupt bacterial resistance mechanisms, thereby increasing the susceptibility of *A. baumannii* to treatment [44–46]. These OMPs represent strategic targets for developing effective vaccines.

In addition to these proteins, the OmpA has been highlighted as another significant vaccine candidate. Research has demonstrated that rOmpA can elicit strong immune responses, and the vaccine's dosage can influence immune polarization, modulating the immunodominant epitopes recognized by T-lymphocytes [47]. Adjuvants, such as MF59, have also been shown to enhance the immunogenicity of fusion proteins derived from *A. baumannii*, suggesting that combining multiple epitopes could generate a more robust immune response [48].

Beyond individual protein-based vaccines, studies have explored alternative approaches, such as using lipopolysaccharide-deficient whole cells as vaccine candidates. These strategies have demonstrated promise in providing protective immunity in experimental models [49]. This approach is especially relevant for *A. baumannii*, which causes severe infections, including pneumonia and bloodstream infections, often with high mortality rates [50].

Moreover, MEVs have gained attention due to their ability to target diverse pathogenic mechanisms of *A. baumannii*, potentially offering broader protection than single-epitope vaccines. The incorporation of vaccine antigens into OMVs has been identified as a promising strategy to improve vaccine efficacy. OMVs enhance antigen delivery and stimulate a stronger immune response, making them a valuable platform for vaccine development against *A. baumannii* [51]. Additionally, synthetic glycoconjugates that mimic the capsular polysaccharides of *A. baumannii* have been proposed as another strategy to leverage the pathogen's virulence factors for protective immunity [52].

In our study, we focused on multi-epitope constructs as promising tools for combating MDR *A. baumannii* infections. Through both *in silico* and *in vivo* approaches, we have designed and evaluated MEVs, providing comprehensive insight into their immunogenic potential and protective efficacy [53,54]. Specifically, we developed an MEV based on antigenic epitopes from the OmpK protein of *A. baumannii*, using computational tools and immunoinformatics to predict and select immunogenic epitopes. These predictions have laid the foundation for innovative vaccine designs with optimized antigenicity and immunogenicity.

*In vivo* studies have further validated the effectiveness of MEVs against *A. baumannii*, demonstrating robust immune responses and protection in animal models [54]. This study proposes an MEV incorporating immunodominant epitopes from OMPs crucial for eliciting sustained immune responses. Immunization with OMPs has been shown to confer high protection levels in both prophylactic and therapeutic animal models [55]. Reverse vaccinology has previously identified surface proteins as promising targets for antibody-based therapies, confirming the potential of novel vaccine candidates [56,57].

Our study utilized epitope shuffling to optimize linear B-cell epitopes for targeting *A. baumannii*, a critical innovation that significantly enhanced the immunogenic properties of the final MEV construct. This technique allowed us to select highly immunogenic combinations, leading to high antigenicity, non-allergenicity, and thermotolerance vaccines. Notably,

the construct was stable against *E. coli*, making it suitable for *in vivo* applications. The selected epitopes derived from four conserved proteins exhibited > 75% prevalence and 100% sequence conservation across diverse *A. baumannii* strains, ensuring robust protection.

Furthermore, our MEV constructs effectively interact with the toll-like receptors TLR2 and TLR4, highlighting its potential to activate critical immune pathways. This feature distinguishes our study from previous studies on limited proteins with narrower antigenicity profiles, such as OmpA and CarO [58]. Our vaccine offers broader protection by prioritizing highly conserved and prevalent proteins such as WP_001019691.1 and WP_000103064.1 and enhancing their antigenic potential through epitope shuffling. Additionally, including protein-protein interaction analyses and TLR binding studies ensures the elicitation of both humoral and cellular immune responses, which are crucial for comprehensive immunity.

Unlike recent efforts that focused on mRNA-based vaccines, our multi-epitope subunit vaccine is more stable and more accessible to produce, offering a viable strategy to combat MDR *A. baumannii* infection [58]. Moreover, the proteins we selected were derived from extracellular loops, enhancing their accessibility to immune recognition and making our approach more effective.

MD simulations conducted in this study provide valuable insights into the structural stability of the MEV, TLR2-MEV, and TLR4-MEV systems. The stable RMSD values of the TLR2-MEV and TLR4-MEV complexes indicate that the interaction of MEV with TLR2 and TLR4 significantly enhances the structural stability of the complexes. This is further supported by the RMSF analysis, which revealed reduced residue fluctuations in the TLR-bound systems, suggesting that these interactions contributed to a more rigid and stable conformation. Consistent Rg values observed in the TLR2-MEV and TLR4-MEV complexes confirm their compactness and stability throughout the simulation. These results suggest that TLR2 and TLR4 stabilize MEVs by reducing large-scale structural deviations and maintaining compactness, which are crucial for these complexes' biological function and efficacy. This enhanced stability may have implications for the design of TLR-targeted therapies or vaccines incorporating MEVs.

The stability of protein-ligand complexes is crucial for understanding their biological functions and therapeutic potential [59]. Recent studies have emphasized using MD simulations and binding affinity analyses to assess stability [60]. MD simulations provide insights into the dynamic behavior of these complexes over time, revealing factors such as RMSD and RMSF that indicate stability. For example, studies on vaccine constructs and antimicrobial peptides (AMPs) have shown that stable interactions, such as consistent hydrogen bonds, contribute to the efficacy of these complexes [61]. Additionally, techniques like MM-PBSA calculations help quantify the binding free energy, further reinforcing the reliability of stability assessments. Hydrogen bonds, hydrophobic interactions, and van der Waals forces are also highlighted as critical factors influencing the overall stability of protein-ligand complexes, with stronger binding typically indicating more stable complexes. In another study focusing on vaccine constructs against carp viruses, molecular docking studies were complemented by MD simulations to evaluate the stability of the constructs when interacting with TLRs. The analysis revealed that vaccine constructs V1 displayed greater stability with TLR3 and TLR5, as evidenced by the RMSD analysis. MD simulations can effectively predict the stability of protein-ligand interactions, providing valuable insights into their potential efficacy in immunological applications [62]. Hydrogen bond analysis further confirmed a stronger binding affinity between the vaccine constructs and TLRs, underscoring the significance of detailed stability assessments in designing effective therapeutic agents.

MEVs for *A. baumannii* target multiple immunodominant epitopes to overcome antigenic diversity, induce antibodies, provide broad protection, and reduce the risk of immune

evasion. Single-antigen vaccines are inadequate because of the diversity of bacteria, necessitating multi-antigen approaches [57]. Effective MEVs require the selection of optimal antigen combinations to induce strong immune responses [63–65]. Research has emphasized the significance of antigen selection, immune protection correlates, and animal models in evaluating efficacy [66]. Promising vaccine candidates have been identified from the outer membrane and extracellular proteins, with immunoinformatics aiding the identification of B- and T-cell epitopes from proteins linked to *A. baumannii* pathogenesis [54,67].

Although computational predictions provide valuable insights into potential vaccine candidates against *A. baumannii*, experimental validation through *in vitro* and *in vivo*, studies are crucial to confirm their biological relevance and therapeutic efficacy. The plan for validation includes *in vitro* assays to assess the antimicrobial potency and safety of drug candidates, such as minimum inhibitory concentration (MIC) and cytotoxicity tests. *In vivo* studies using animal models will evaluate the immune response and protective efficacy of vaccine candidates, as well as the pharmacokinetics and toxicology of the candidates. Immunological assessments will use ELISA and flow cytometry to measure immune responses, with thorough data analysis performed to validate the computational predictions.

Studies have highlighted the importance of transitioning from computational predictions to experimental verification. For instance, Moein et al. (2024) [68] identified antifungal plant flavonoids through *in silico* methods. They later validated their ability to control rice blast disease caused by *Magnaporthe oryzae* through experimental testing. Ong et al. (2021) [69] also utilized the Vaxign2 framework to predict vaccine antigen targets for various pathogens, including *A. baumannii*. They emphasized the necessity of experimental validation to confirm the effectiveness and safety of the expected vaccine candidates.

Similarly, Aiman et al. (2023) highlighted using vaccinomics and immunoinformatics to design multi-epitope chimeric vaccines against multiple pathogens, including *A. baumannii* [70]. They noted that these constructs were validated experimentally, reinforcing the importance of such approaches in vaccine development. Moreover, Khan et al. (2018) [71] demonstrated the computational identification and validation of potential antigenic peptide vaccines, emphasizing the need for experimental confirmation of predicted T-cell epitopes to ensure long-lasting immune responses. These examples underscore the necessity of bridging the gap between computational predictions and experimental validation to ensure that the vaccine candidates are effective and safe for potential therapeutic development.

The results of this study demonstrate that the designed vaccine elicits a robust and multifaceted immune response, effectively activating both humoral and cellular immunity. Early-phase humoral immunity is marked by activating IgM-expressing B cells, initiating a primary immune response, and the production of high levels of IgG antibodies, establishing long-term immunity. The vaccine also triggers a strong Th1-dominated cell-mediated response, evidenced by the significant increase in Th1 cell populations and elevated IFN-γ levels. IFN-γ, a key cytokine involved in macrophage activation and intracellular pathogen clearance, highlights the vaccine's ability to stimulate potent cellular immunity. Elevated IL-2 levels further underscore an active inflammatory response, which is critical for initiating and amplifying the immune reaction. Consistent activity of innate immune cells, such as macrophages, dendritic cells, and NK cells, further supports adaptive immune functions. Distinct cytokine patterns, including early peaks of IL-2 and TNF-α and sustained IFN-γ activity, underscore the tightly regulated immune signaling required for effective immune modulation. These findings collectively highlight the vaccine's capacity to provide immediate and durable protection through coordinated activation of humoral and cellular immune pathways. The findings of this study are consistent with previous research on immune activation in response to bacterial vaccines, particularly those targeting *A. baumannii*. For example, Jeffreys et al.

(2022) demonstrated that vaccination with a recombinant protein from *A. baumannii* induced a robust IgM response, which was associated with protective immunity against subsequent pathogen challenges [72]. This aligns with the current study's observation of a significant elevation in IgM-expressing B-cells, indicating a primary immune response initiation. Further supporting the current findings, Chen et al. (2020) emphasized the importance of Th1 responses in combating *A. baumannii* infections, showing that IFN-γ production was critical for macrophage activation and bacterial clearance [73]. The substantial increase in Th1 cell populations and elevated IFN-γ levels in our study corroborate these findings, suggesting a similar protective mechanism at play. However, while these results are promising, further studies are needed to assess the durability of this immune response. Specifically, it is crucial to evaluate IgG class switching, memory B and T cell formation, and the long-term maintenance of immunity. Research by Yates (2013) [74] highlighted the importance of these processes in establishing long-term immunity, a point underscored by Hjalmsdottir et al. (2024) [75]. The next step involves designing and conducting experimental studies to validate the immunogenicity and safety profile of the proposed vaccine.

Although this study presents promising vaccine candidate targets through computational predictions, several limitations must be acknowledged. First, reliance on *in silico* methods means that immunogenicity, antigenicity, and vaccine efficacy predictions require extensive experimental validation. Although computational predictions are insightful, they cannot fully replicate the complexities of host-pathogen interactions or immune responses in a biological system. Therefore, it is beneficial for the effectiveness of MEV to be confirmed through *in vitro* and *in vivo* studies.

Another limitation is the potential variability in *A. baumannii* strains. Although MEV targets highly conserved epitopes, there remains a risk of antigenic variation across different clinical isolates, which may affect the vaccine's efficacy. Lastly, this stage's absence of experimental validation limits the study's immediate translational impact. Further investigations into the safety, immunogenicity, and therapeutic efficacy of the proposed candidates will be beneficial in confirming their real-world applicability.

## 5. Conclusion

This study underscores the urgent need for innovative therapeutic strategies against *A. baumannii*, a formidable pathogen due to the rapid development of MDR strains. No approved vaccines are available, highlighting the challenges in developing effective solutions. Advances in vaccine research, particularly in the development of MEVs targeting OMPs, present promising avenues for combating infections. This study successfully employed *in silico* methods to identify and optimize immunogenic epitopes, leading to an MEV candidate with high antigenicity and stability that can activate immune responses. MEV targets highly conserved epitopes across diverse strains, ensuring broad protection. Molecular docking studies, supported by MD simulations, highlighted the vaccine's capacity to form strong and stable interactions with immune receptors.

## Supporting information

**S1 Table. Characterization of 138 extracellular and outer membrane proteins of *A. baumannii* strain VB7036.**
(DOCX)

**S2 Table. Epitope Shuffling Techniques for Optimal Arrangement with Highest Antigenicity Score.**
(DOCX)

**S1 Fig. ProSA-web analysis of eight multi-epitope constructs.**
(DOCX)

AcknowledgmentThe authors express their heartfelt gratitude to the Pasteur Institute of Iran staff for their unwavering moral support.

## Author contributions

**Conceptualization:** Masoumeh Beig, Farzad Badmasti.

**Data curation:** Masoumeh Beig, Farzad Badmasti.

**Formal analysis:** Masoumeh Beig, Safoura Moradkasani.

**Methodology:** Mohammad Sholeh, Safoura Moradkasani, Behzad Shahbazi.

**Project administration:** Farzad Badmasti.

**Software:** Mohammad Sholeh, Farzad Badmasti.

**Validation:** Masoumeh Beig, Mohammad Sholeh.

**Visualization:** Safoura Moradkasani.

**Writing – original draft:** Masoumeh Beig, Behzad Shahbazi.

**Writing – review & editing:** Masoumeh Beig, Mohammad Sholeh, Safoura Moradkasani, Farzad Badmasti.

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
