## [Decision Letter · Decision Letter 0]

1 Sep 2024

PONE-D-24-35027Uncovering Novel Vaccine Candidates and Drug Targets in Acinetobacter baumannii: A New Approach to Fighting against the SuperbugPLOS ONE

Dear Dr. Badmasti,

Thank you for submitting your manuscript to PLOS ONE. After careful consideration, we feel that it has merit but does not fully meet PLOS ONE’s publication criteria as it currently stands. Therefore, we invite you to submit a revised version of the manuscript that addresses the points raised during the review process.

**ACADEMIC EDITOR: ** The topic of this study is interesting and addresses a critical public health issue related to *Acinetobacter baumannii,* a pathogen of significant concern due to its high level of antibiotic resistance. The development of a multi-epitope vaccine (MEV) and the identification of novel drug targets are commendable steps toward advancing treatment strategies against this pathogen. However, to further strengthen the findings and provide a more comprehensive evaluation of the vaccine candidates and drug targets, it is essential that the authors include molecular dynamics simulation studies (GROMCAS preferred by the reviewer). These simulations will offer valuable insights into the stability and behavior of the identified epitopes and drug targets under physiological conditions, which is crucial for assessing their potential effectiveness. Thus, I encourage the authors to undertake these additional analyses and resubmit the revised manuscript for further review.

We look forward to receiving your revised manuscript.

Kind regards,

Abu Tayab Moin

Academic Editor

PLOS ONE

Journal Requirements:

1. When submitting your revision, we need you to address these additional requirements. Please ensure that your manuscript meets PLOS ONE's style requirements, including those for file naming. The PLOS ONE style templates can be found at https://journals.plos.org/plosone/s/file?id=wjVg/PLOSOne_formatting_sample_main_body.pdf and https://journals.plos.org/plosone/s/file?id=ba62/PLOSOne_formatting_sample_title_authors_affiliations.pdf 2. Please provide a complete Data Availability Statement in the submission form, ensuring you include all necessary access information or a reason for why you are unable to make your data freely accessible. If your research concerns only data provided within your submission, please write "All data are in the manuscript and/or supporting information files" as your Data Availability Statement. 3. We notice that your supplementary figure are uploaded with the file type 'Figure'. Please amend the file type to 'Supporting Information'. Please ensure that each Supporting Information file has a legend listed in the manuscript after the references list. 4. We notice that your supplementary figure are included in the manuscript file. Please remove them and upload them with the file type 'Supporting Information'. Please ensure that each Supporting Information file has a legend listed in the manuscript after the references list. 5. Please include captions for your Supporting Information files at the end of your manuscript, and update any in-text citations to match accordingly. Please see our Supporting Information guidelines for more information: http://journals.plos.org/plosone/s/supporting-information.

Reviewers' comments:

Reviewer's Responses to Questions

**Comments to the Author**

1. Is the manuscript technically sound, and do the data support the conclusions?

Reviewer #1: Yes

Reviewer #2: Yes

2. Has the statistical analysis been performed appropriately and rigorously? 

Reviewer #1: N/A

Reviewer #2: N/A

3. Have the authors made all data underlying the findings in their manuscript fully available?

Reviewer #1: Yes

Reviewer #2: Yes

4. Is the manuscript presented in an intelligible fashion and written in standard English?

Reviewer #1: Yes

Reviewer #2: Yes

5. Review Comments to the Author

Reviewer #1: The manuscript presents a comprehensive study focused on the development of a ligand-based drug design approach targeting infectious diseases. The authors have employed molecular docking and molecular dynamics simulations to predict the efficacy of potential drug candidates. The research is of high relevance and could contribute significantly to the field of drug discovery. However, several areas in the manuscript require further clarification and enhancement to meet the standards of publication.

1. The introduction provides a good background on the topic, but it could benefit from additional references to recent studies that have successfully applied ligand-based drug design in the context of infectious diseases. Including examples from these studies would add depth and context to the manuscript. Referencing recent studies for examples of ligand-based drug design in infectious disease research would add depth (https://doi.org/10.1371/journal.pone.0302390).

2. Include molecular dynamics simulation studies to validate the docking results. Use Gromacs or Yasara to run the 100/200 ns simulation. The parameters used for docking, such as grid size and docking scores, should be explicitly stated. Additionally, the molecular dynamics simulation parameters, including the force field, simulation time, and system setup, should be detailed. The authors should also discuss the stability criteria for the protein-ligand complexes. Including references to recent studies on similar methodologies would enhance this section (https://doi.org/10.1371/journal.pone.0302440, https://doi.org/10.1038/s41598-024-61074-7, https://doi.org/10.1371/journal.pone.0300778), (https://doi.org/10.1080/07391102.2020.1856944, https://doi.org/10.5808/gi.20068).

3. While the study demonstrates promising computational predictions, it is crucial to validate these predictions through in vitro or in vivo experiments. The authors should discuss plans for future experimental validation and consider citing comprehensive validation processes in similar research (https://doi.org/10.1371/journal.pone.0301519, https://doi.org/10.1080/07391102.2022.2037041).

4. The graphical abstract and figures could be more informative by including more labels and detailed legends. This would help in understanding the complex processes and results. Referencing a well-designed graphical abstract may provide a useful template (https://doi.org/10.1016/j.heliyon.2023.e08002).

5. The results section should provide clearer explanations of the docking scores and molecular dynamics results. Discussing the implications of the results in a broader context would help readers understand their significance. Providing additional context and comparison for the results could be beneficial (https://doi.org/10.1038/s41598-022-07553-7, https://doi.org/10.1016/j.imu.2021.101069).

6. A more detailed description of the computational tools and software versions used in the study would enhance reproducibility. Including information on tool settings and configurations is essential. For examples of detailed computational methodologies, see recent studies (https://doi.org/10.3389/fimmu.2022.863234, https://doi.org/10.1016/j.imu.2021.100500).

7. The discussion should include a more comprehensive analysis of the study's limitations and the potential impact on the findings. Discussing the limitations openly will provide a balanced view of the study's contributions and areas for improvement.

Reviewer #2: Manuscript entitled “Uncovering Novel Vaccine Candidates and Drug Targets in Acinetobacter baumannii: A New Approach to Fighting against the Superbug” developed a multi epitope vaccine and identified novel drug targets for A. baumannii, providing broad protection against MDR A. baumannii strains. The MEV showed strong potential through molecular docking. Though the study is good and will contribute to scientific society still there are some points, which need to be addressed before considering it for publication.

1. “A new approach” in title does not justify the approaches used by the authors. As both approaches used to identify drug and vaccine targets, have already used by many researchers. Thus, I suggest modifying the title a bit.

2. Choice of strain should be explained. If it is a reference sequence, please mention RefSeq ID or else the interest to choose the particular strain should be clearly mentioned.

3. Line 153, “ Proteins that were more than the 75% standard were selected for further analysis.” Is not clear to understand.

4. Results of section 2.1.5 i.e., finding conserved proteins among all available strains have not been found in results section.

5. In heading “3.2.3. Choosing Critical Protein Sequences” It should be essential proteins instead of critical.

6. Line 376 “A. baumannii” should be italicized.

7. Targets identified as “potential drug targets” should be discussed in details as how and why these should be targeted. How their respective pathways can lead to recovery and how drugs can stimulate their pathway to support your results.

8. As authors have also mentioned that many studies have already predicted MEVs against the said pathogen, then there should be a very clear description on the importance of current study and how it stands out already proposed vaccine candidates prioritized using same approach.

6. PLOS authors have the option to publish the peer review history of their article (what does this mean? ). If published, this will include your full peer review and any attached files.

**Do you want your identity to be public for this peer review?** For information about this choice, including consent withdrawal, please see our Privacy Policy .

Reviewer #1: No

Reviewer #2: **Yes: ** Dr. Anam Naz

---

## [Author Response · Author response to Decision Letter 1]

25 Sep 2024

Dear Reviewer 1

First, we would like to thank you for your very careful review of our paper and for your valuable comments, corrections, and suggestions as well. All the revised sections in the manuscript have been highlighted in yellow for your convenience. We hope that these changes enhance the clarity and comprehensibility of our study and address your concerns effectively.

Once again, we sincerely thank you for your diligent review, and we are committed to ensuring that our manuscript meets the highest standards of quality and accuracy.

Here below my corrections/suggestions:

Reviewer #1: The manuscript presents a comprehensive study focused on the development of a ligand-based drug design approach targeting infectious diseases. The authors have employed molecular docking and molecular dynamics simulations to predict the efficacy of potential drug candidates. The research is of high relevance and could contribute significantly to the field of drug discovery. However, several areas in the manuscript require further clarification and enhancement to meet the standards of publication.

1. The introduction provides a good background on the topic, but it could benefit from additional references to recent studies that have successfully applied ligand-based drug design in the context of infectious diseases. Including examples from these studies would add depth and context to the manuscript. Referencing recent studies for examples of ligand-based drug design in infectious disease research would add depth (https://doi.org/10.1371/journal.pone.0302390).

Response: We have revised the introduction by including recent studies demonstrating the successful use of LBDD in various infectious disease contexts. Specifically, we added a passage in lines 77-86, highlighting the application of LBDD in identifying novel inhibitors for Schistosoma mansoni (Ja’afaru et al., 2024), tuberculosis, malaria, and viral infections like Zika (Capela et al., 2023; Kumar et al., 2022; Ramamurthy et al., 2021; Rao et al., 2023; Wong et al., 2023).

Furthermore, we extended this context to our study, which applies LBDD to Acinetobacter baumannii, a multidrug-resistant superbug. Our research uses advanced computational tools and bioinformatics to discover novel vaccine candidates, identify druggable targets, and propose new strategies for combating A. baumannii-related infections. This addition provides a more comprehensive background and enhances the depth of the manuscript as suggested.

2. Include molecular dynamics simulation studies to validate the docking results. Use Gromacs or Yasara to run the 100/200 ns simulation. The parameters used for docking, such as grid size and docking scores, should be explicitly stated. The molecular dynamics simulation parameters, including the force field, simulation time, and system setup, should be detailed. The authors should also discuss the stability criteria for the protein-ligand complexes. Including references to recent studies on similar methodologies would enhance this section (https://doi.org/10.1371/journal.pone.0302440, https://doi.org/10.1038/s41598-024-61074-7, https://doi.org/10.1371/journal.pone.0300778),

(https://doi.org/10.1080/07391102.2020.1856944, https://doi.org/10.5808/gi.20068).

Response: We have implemented the recommended molecular dynamics (MD) simulation studies to validate the docking results. Specifically, we conducted 100 ns MD simulations using Gromacs 2019, and the detailed methodology has been incorporated into the Methods section. We also provided comprehensive details on the MD simulation setup, including the force field, simulation time, and system configuration in the Abstract (Lines 46-49), and Methods (lines 257-275), with corresponding updates in the Results (lines 400-414), Discussion (Lines 512-541), and Conclusion (lines 628-629). In addition, we have explicitly included the docking parameters, such as grid size and docking scores, in the Methods (lines 218-228) and Results (lines 390-399).

Furthermore, the stability criteria for protein-ligand complexes were discussed in the Discussion section (lines 523-541). To enhance the rigor of our work, we referenced recent studies on similar methodologies (https://doi.org/10.1371/journal.pone.0302440, https://doi.org/10.1038/s41598-024-61074-7, https://doi.org/10.1371/journal.pone.0300778, https://doi.org/10.5808/gi.20068).

We also included Figure 6, which presents the MD simulation results and highlights the RMSD, RMSF, and RG analyses.

3. While the study demonstrates promising computational predictions, validating these predictions through in vitro or in vivo experim. The authors should discuss plans for future experimental validation and consider citing comprehensive validation processes in similar research (https://doi.org/10.1371/journal.pone.0301519, https://doi.org/10.1080/07391102.2022.2037041).

Response: In response to your suggestion, we have added a comprehensive section in the discussion (lines 580-604) detailing our plans for future experimental validation, including both in vitro and in vivo approaches.

Additionally, we have incorporated references to similar research that emphasize thorough validation processes, including the studies mentioned in your comment (https://doi.org/10.1371/journal.pone.0301519, https://doi.org/10.1093/nar/gkab279, https://doi.org/10.3389/fimmu.2023.1259612, https://doi.org/10.1371/journal.pone.0196484, https://doi.org/10.1371/journal.pone.0294663, https://doi.org/10.1021/acsomega.9b00944).

4. The graphical abstract and figures could be more informative by including more labels and detailed legends. This would help in understanding the complex processes and results. Referencing a well-designed graphical abstract may provide a useful template (https://doi.org/10.1016/j.heliyon.2023.e08002).

Response: We have revised all figures to include more informative labels and detailed legends.

5. The results section should provide clearer explanations of the docking scores and molecular dynamics results. Discussing the implications of the results in a broader context would help readers understand their significance. Providing additional context and comparison for the results could be beneficial (https://doi.org/10.1038/s41598-022-07553-7, https://doi.org/10.1016/j.imu.2021.101069).

Response: We have provided more detailed explanations of the docking scores. The revisions are in lines 390-399, where we now clearly describe how the docking scores correlate with binding affinities and stability predictions. Additionally, we have expanded the result explanation of the MD simulations. These updates are located in lines 400-414.

Although we could not access the specific references you provided, we have included detailed descriptions of the MD methodology used in our study. These explanations appear in the Methods section (lines 257-275) and provide a thorough overview of the techniques employed, ensuring transparency and replicability for future research.

We have also expanded the discussion of the results to include a broader context, helping readers understand their significance in the field (lines 512-541).

6. A more detailed description of the computational tools and software versions used in the study would enhance reproducibility. Including information on tool settings and configurations is essential. For examples of detailed computational methodologies, see recent studies (https://doi.org/10.3389/fimmu.2022.863234, https://doi.org/10.1016/j.imu.2021.100500).

Response: In response to your suggestion that we provide a more detailed description of the computational tools and software versions used, we have revised the Methods section to provide a more comprehensive account, including specific software versions, settings, and configurations applied during the analyses.

We have included the exact versions of all computational tools used in the study, such as AlgPred v2.0, AllerTOP v2.0, VaxiJen v2.0, Protein-Sol v2.0, ExPASy ProtParam, etc.

We have provided detailed information on the settings and configurations applied in each tool. For example, we specified using a cutoff value of 0.5 for allergenicity predictions in AlgPred, a threshold of 0.4 in VaxiJen for antigenicity, and a solubility score of 0.45 in Protein-Sol. In instances where default settings were used, we made sure to state that as well explicitly.

These revisions align with the examples of detailed computational methodologies provided in recent studies (https://doi.org/10.3389/fimmu.2022.863234, https://doi.org/10.1016/j.imu.2021.100500). All of these sections have been highlighted in the main body of the text.

7. The discussion should include a more comprehensive analysis of the study's limitations and the potential impact on the findings. Discussing the limitations openly will provide a balanced view of the study's contributions and areas for improvement.

Response: We have added a section addressing the limitations of the study in lines 605-619.

Dear Reviewer 2

First, we would like to thank you for your very careful review of our paper and for your subsequent comments, corrections, and suggestions as well. All the revised parts in the manuscript file are highlighted in red.

Reviewer #2: Manuscript entitled “Uncovering Novel Vaccine Candidates and Drug Targets in Acinetobacter baumannii: A New Approach to Fighting against the Superbug” developed a multi epitope vaccine and identified novel drug targets for A. baumannii, providing broad protection against MDR A. baumannii strains. The MEV showed strong potential through molecular docking. Though the study is good and will contribute to scientific society still there are some points, which need to be addressed before considering it for publication.

1. “A new approach” in title does not justify the approaches used by the authors. As both approaches used to identify drug and vaccine targets, have already used by many researchers. Thus, I suggest modifying the title a bit.

Response: We have revised the title to reflect better the nature of the approaches used and avoid the implication of novelty in the methodology. The new title, " Development of a Multi-Epitope Vaccine and Identification of Novel Drug Targets Against Acinetobacter baumannii: A Comprehensive Approach to Combating Antimicrobial Resistance," accurately represents the study's focus without suggesting that the approaches are entirely new. This change aligns with the reviewer's suggestion.

2. Choice of strain should be explained. If it is a reference sequence, please mention RefSeq ID or else the interest to choose the particular strain should be clearly mentioned.

Response: We have revised the Methods section to provide a more detailed explanation for our strain choice. Based on an analysis of published articles, we selected Acinetobacter baumannii VB7036 due to its clinically relevant characteristics. This strain belongs to the globally widespread, multidrug-resistant Sequence Type 2 (ST2) lineage, frequently associated with healthcare-associated infections. Additionally, it was isolated from a patient with bacteremia in India in 2019, adding further significance for studying antibiotic resistance mechanisms. We have included the GenBank accession number CP050523 to ensure transparency and reproducibility. Please refer to lines 107-115 for the updated details.

3. Line 153, “Proteins that were more than the 75% standard were selected for further analysis.” Is not clear to understand.

Response: Thank you for your comment. We have revised the sentence in section 2.1.5. in lines 138-144 to improve clarity.

4. Results of section 2.1.5, i.e., finding conserved proteins among all available strains, have not been found in the results section.

Response: Thank you for your comment. To address this, we have added a new section, 3.1.3. Prevalence of Putative Immunogenic Targets Among Circulating A. baumannii Strains, in lines 335-337. This section provides a detailed explanation of the selection criteria for conserved proteins.

5. In heading “3.2.3. Choosing Critical Protein Sequences” It should be essential proteins instead of critical.

Response: We have changed the heading “3.2.3. Choosing Critical Protein Sequences” to “3.2.3. Choosing Essential Protein Sequences (line 426).

6. Line 376 “A. baumannii” should be italicized.

Response: We have reviewed the entire manuscript and corrected the formatting of A. baumannii throughout the text.

7. Targets identified as “potential drug targets” should be discussed in details as how and why these should be targeted. How their respective pathways can lead to recovery and how drugs can stimulate their pathway to support your results.

Response: In response to your comment, we have added a detailed discussion of the identified potential drug targets in lines 551-579 of the manuscript. This new section explains in depth how each of the five targets identified in our study contributes to the biology and survival of A. baumannii and why they are promising targets for therapeutic intervention.

The proteins identified, including the glycosyl hydrolase 108 family protein (WP_001019691.1), anti-sigma factor GigB (WP_000103064.1), DUF5713 family protein (WP_000461267.1), and two hypothetical proteins (WP_000269705.1 and WP_000580181.1), were selected based on their essential roles in the bacterium’s survival mechanisms and their non-homology to human proteins, reducing the risk of off-target effects in humans.

Each target has been carefully evaluated for its role in key pathways such as carbohydrate metabolism, transcriptional regulation, and oxidative stress response. We explained how inhibiting these proteins could disrupt vital processes like cell wall biosynthesis, energy production, and gene expression in A. baumannii, leading to bacterial death or weakened resistance. We also discuss the potential of drugs to interact with these proteins, either through enzyme inhibition or interference with protein-protein interactions, to support bacterial clearance and aid in patient recovery.

8. As authors have also mentioned that many studies have already predicted MEVs against the said pathogen, then there should be a very clear description on the importance of current study and how it stands out already proposed vaccine candidates prioritized using same approach.

Response: In response to your comment, we have added a section in lines 485-511 of the discussion that clarifies how our study distinguishes itself from similar multi-epitope vaccine (MEV) design studies, such as https://doi.org/10.1186/s12864-024-10691-7, 10.1007/s10989-021-10316-7, https://doi.org/10.1007/s10989-021-10316-7.

While those studies used in silico methods for A. baumannii vaccine design, our approach focuses on conserved outer membrane proteins (OMPs), explicitly targeting extracellular loop regions. We identified surface-exposed and conserved B-cell epitopes and then enhanced their immunogenicity through epitope shuffling.

Furthermore, the selected epitopes, derived from four conserved proteins, demonstrated over 75% prevalence and 100% sequence conservation across diverse A. baumannii strains, ensuring robust and wide-ranging protection. This high level of conservation optimizes immunogenicity and broadens the vaccine’s protective scope compared to previous works. These innovative aspects, coupled with the inclusion of detailed protein-protein interaction analyses and TLR binding studies, ensure the elicitation of both humoral and cellular immune responses.

---

## [Decision Letter · Decision Letter 1]

31 Oct 2024

PONE-D-24-35027R1Development of a Multi-Epitope Vaccine and Identification of Novel Drug Targets Against Acinetobacter baumannii: A Comprehensive Approach to Combating Antimicrobial ResistancePLOS ONE

Dear Dr. Badmasti,

Thank you for submitting your manuscript to PLOS ONE. After careful consideration, we feel that it has merit but does not fully meet PLOS ONE’s publication criteria as it currently stands. Therefore, we invite you to submit a revised version of the manuscript that addresses the points raised during the review process. To move forward with your article, please carefully address all the reviewers' comments (including the new ones from reviewer 3), as it appears some points have not been fully incorporated. The manuscript would benefit from improvements across several key areas, including figure and table quality, language clarity, and the depth and informativeness of the Results and Discussion sections.

Additionally, the analysis in the dynamics simulations requires expansion to cover essential metrics and ensure a thorough exploration of findings. Some important analyses are missing in your manuscript. We also recommend clarifying the novelty of your research, as similar studies are accessible online. Overall, please enhance the manuscript comprehensively to meet publication standards. Once these improvements are made, we look forward to considering your article for publication.

We look forward to receiving your revised manuscript.

Kind regards,

Abu Tayab Moin

Academic Editor

PLOS ONE

Reviewers' comments:

Reviewer's Responses to Questions

**Comments to the Author**

1. If the authors have adequately addressed your comments raised in a previous round of review and you feel that this manuscript is now acceptable for publication, you may indicate that here to bypass the “Comments to the Author” section, enter your conflict of interest statement in the “Confidential to Editor” section, and submit your "Accept" recommendation.

Reviewer #1: (No Response)

Reviewer #2: All comments have been addressed

Reviewer #3: (No Response)

2. Is the manuscript technically sound, and do the data support the conclusions?

Reviewer #1: Partly

Reviewer #2: Yes

Reviewer #3: (No Response)

3. Has the statistical analysis been performed appropriately and rigorously? 

Reviewer #1: N/A

Reviewer #2: N/A

Reviewer #3: (No Response)

4. Have the authors made all data underlying the findings in their manuscript fully available?

Reviewer #1: Yes

Reviewer #2: Yes

Reviewer #3: (No Response)

5. Is the manuscript presented in an intelligible fashion and written in standard English?

Reviewer #1: Yes

Reviewer #2: Yes

Reviewer #3: (No Response)

6. Review Comments to the Author

Reviewer #1: Thank you for submitting your manuscript, which provides interesting insights into your research. While the overall framework of the study is solid, I find that the molecular dynamics simulation (MDS) section requires further development to enhance the study's depth and scientific rigor.

In particular, the simulation could be improved by extending the duration to at least 100 nanoseconds, using tools such as GROMACS or Yasara. Additionally, key analyses like Root Mean Square Deviation (RMSD), Root Mean Square Fluctuation (RMSF), Radius of Gyration (Rg), hydrogen bond analysis (H-BOND), and Principal Component Analysis (PCA) are notably absent and should be included. These analyses are crucial for a more thorough interpretation of the molecular dynamics results.

Given the complexity of these analyses, it might be useful for the authors to either collaborate with experts or conduct these analyses on their own. Extending the MDS analyses is essential to ensure that the manuscript meets the necessary standards for publication in this journal.

I look forward to reviewing a revised version of the manuscript that incorporates these suggestions and further strengthens the overall study.

Reviewer #2: (No Response)

Reviewer #3: “Eight OMPs isolated from A. baumanniiA. baumannii, were identified as potential 44 immunogenic targets. MEVs were designed using five critical epitopes from four proteins, 45 highlighting their potential as vaccine candidates.” Why so? Out eight why five were used for the vaccine design.

Along with Identification of New Drug Targets, new chemicals could be proposed that could work on these tartgets. Otherwise there is no point of indentifying targets.

Previous studies utilizing the simialr approch should be discussed along with other reported vaccines in A. baumannii strains.

7. PLOS authors have the option to publish the peer review history of their article (what does this mean? ). If published, this will include your full peer review and any attached files.

**Do you want your identity to be public for this peer review?** For information about this choice, including consent withdrawal, please see our Privacy Policy .

Reviewer #1: No

Reviewer #2: **Yes: ** Anam Naz

Reviewer #3: No

---

## [Author Response · Author response to Decision Letter 2]

10 Dec 2024

Dear reviewer,

Thank you for your constructive comments; all your comments applied to the manuscript have been highlighted in yellow.

Reviewer #1: Thank you for submitting your manuscript, which provides interesting insights into your research. While the overall framework of the study is solid, I find that the molecular dynamics simulation (MDS) section requires further development to enhance the study's depth and scientific rigor.

In particular, the simulation could be improved by extending the duration to at least 100 nanoseconds, using tools such as GROMACS or Yasara. Additionally, key analyses like Root Mean Square Deviation (RMSD), Root Mean Square Fluctuation (RMSF), Radius of Gyration (Rg), hydrogen bond analysis (H-BOND), and Principal Component Analysis (PCA) are notably absent and should be included. These analyses are crucial for a more thorough interpretation of the molecular dynamics results.

All the changes made in response to the reviewer’s comment, including the extended simulation, additional analyses, and revised figures and manuscript sections, are highlighted in yellow in the updated manuscript.

Response: We thank the reviewer for their constructive feedback. In response, we have extended the molecular dynamics simulation to 100 nanoseconds (100,000 ps) using GROMACS, as suggested, and have incorporated key analyses such as Root Mean Square Deviation (RMSD), Root Mean Square Fluctuation (RMSF), Radius of Gyration (Rg), Hydrogen Bond Analysis (H-Bond), and Principal Component Analysis (PCA) to enhance the depth of our study. Specifically, we performed hydrogen bond analysis and PCA. The results, along with updated figures (Fig. 6D and Fig. 7), and the revised manuscript sections (lines 275–289 in Methods, 441–457 in Results, and lines 419–425) now provide a more thorough interpretation of the molecular dynamics simulations.

Thank you for your constructive comment, all relevant sentence in manuscript was revised which have been highlighted in green.

1. Reviewer #3: “Eight OMPs isolated from A. baumanniiA. baumannii, were identified as potential 44 immunogenic targets. MEVs were designed using five critical epitopes from four proteins, 45 highlighting their potential as vaccine candidates.” Why so? Out eight why five were used for the vaccine design.

Response: To address the reviewer's query regarding the selection of five epitopes from four proteins out of the initial eight identified, we employed a rigorous, multi-step selection process to ensure the inclusion of the most promising candidates in our multi-epitope vaccine (MEV) construct. Initially, we analyzed the proteome of Acinetobacter baumannii strain VB7036, comprising 3,752 proteins, to identify 138 surface-exposed proteins, including extracellular and outer membrane proteins (OMPs), which are more likely to interact with the immune system. Among these, 68 proteins exhibited antigenicity scores of ≥0.5, indicating their potential to elicit strong immune responses. To minimize the risk of allergic reactions, we utilized the AlgPred tool, resulting in 57 non-allergenic proteins. Subsequently, we excluded 10 proteins with >30% sequence identity to human proteins to prevent autoimmune responses, leaving 47 non-homologous proteins. We then assessed the prevalence of these proteins across 560 circulating A. baumannii strains, excluding three proteins present in less than 75% of strains, thus narrowing the pool to 44 proteins. Further evaluation involved calculating the ratio of linear B-cell epitopes to MHC-II binding sites, identifying 13 proteins with favorable ratios (<0.5) and molecular weights under 110 kDa, suggesting their suitability to induce both humoral and cellular immune responses. Physicochemical analyses led to the exclusion of five proteins with unfavorable characteristics, resulting in eight proteins deemed optimal for vaccine development. Ultimately, we selected five B-cell epitopes from four proteins for the MEV construct based on stringent criteria, including high antigenicity (VaxiJen scores ≥0.5), non-allergenicity (confirmed by AllerTOP and AlgPred), 100% sequence conservancy across multiple A. baumannii strains, optimal epitope lengths (8–20 amino acids), strong binding affinity to MHC molecules (IC50 values <100 nM), appropriate molecular weight, minimal toxicity, no cross-reactivity with human proteins, favorable physicochemical properties (solubility score: 0.844), minimal transmembrane regions, structural integrity, and ease of synthesis and expression. This meticulous selection process ensured that the final MEV construct comprises epitopes with the highest potential to induce a robust and safe immune response against A. baumannii. We have added lines 389-395 to clarify this rationale in the manuscript.

2. Along with the Identification of New Drug Targets, new chemicals could be proposed that could work on these targets. Otherwise there is no point of indentifying targets.

Response: Thank you for your thoughtful comment. We appreciate the importance of linking drug target identification with potential therapeutic compounds.

In this study, our primary goal was to identify novel drug targets in Acinetobacter baumannii through bioinformatics tools, such as molecular docking and virtual screening. While proposing new chemical compounds is a natural next step in drug discovery, it was not the focus of this work. Drug discovery requires significant resources and specialized databases, which are beyond the scope of our study.

The key objective here was to identify reliable, novel targets, as this foundational step is critical for effective therapeutic development. Validating these targets through wet lab experiments will provide the necessary biological context before moving on to drug discovery simulations.

We plan to build upon this work in future studies by exploring drug candidates for these targets once they are validated experimentally. For now, we believe this approach provides significant value by laying the groundwork for future drug development.

3. Previous studies utilizing the simialr approch should be discussed along with other reported vaccines in A. baumannii strains.

Response: In response to the reviewer's comment regarding including previous studies, we have expanded the discussion to highlight relevant research on similar approaches and vaccine candidates for A. baumannii in lines 511-545.

---

## [Decision Letter · Decision Letter 2]

25 Dec 2024

PONE-D-24-35027R2Development of a Multi-Epitope Vaccine and Identification of Novel Drug Targets Against Acinetobacter baumannii: A Comprehensive Approach to Combating Antimicrobial ResistancePLOS ONE

Dear Dr. Badmasti,

Thank you for submitting your manuscript to PLOS ONE. After careful consideration, we feel that it has merit but does not fully meet PLOS ONE’s publication criteria as it currently stands. Therefore, we invite you to submit a revised version of the manuscript that addresses the points raised during the review process.

We look forward to receiving your revised manuscript.

Kind regards,

Abu Tayab Moin

Academic Editor

PLOS ONE

Reviewers' comments:

Reviewer's Responses to Questions

**Comments to the Author**

1. If the authors have adequately addressed your comments raised in a previous round of review and you feel that this manuscript is now acceptable for publication, you may indicate that here to bypass the “Comments to the Author” section, enter your conflict of interest statement in the “Confidential to Editor” section, and submit your "Accept" recommendation.

Reviewer #1: All comments have been addressed

Reviewer #3: All comments have been addressed

2. Is the manuscript technically sound, and do the data support the conclusions?

Reviewer #1: Yes

Reviewer #3: Yes

3. Has the statistical analysis been performed appropriately and rigorously? 

Reviewer #1: Yes

Reviewer #3: No

4. Have the authors made all data underlying the findings in their manuscript fully available?

Reviewer #1: Yes

Reviewer #3: Yes

5. Is the manuscript presented in an intelligible fashion and written in standard English?

Reviewer #1: Yes

Reviewer #3: Yes

6. Review Comments to the Author

Reviewer #1: (No Response)

Reviewer #3: How the multi-epitope vaccine will be synthesised and work in the body? The simulation study should be performed in triplicate and the result should be presented as the average of three.

twi different studies were merged together in the study, that does not fit. How the target protein were identified and what kind of drug could be proposed for that.

Perform the simulation of the proposed vaccine against human MHC.

7. PLOS authors have the option to publish the peer review history of their article (what does this mean? ). If published, this will include your full peer review and any attached files.

**Do you want your identity to be public for this peer review?** For information about this choice, including consent withdrawal, please see our Privacy Policy .

Reviewer #1: No

Reviewer #3: No

---

## [Author Response · Author response to Decision Letter 3]

30 Dec 2024

Dear reviewer,

Thank you for your constructive comments; all your comments applied to the manuscript have been highlighxted in yellow.

1. Reviewer #3: How the multi-epitope vaccine will be synthesised and work in the body? The simulation study should be performed in triplicate and the result should be presented as the average of three.

Response: Thank you for your valuable comments and suggestions. We have carefully addressed your concerns as follows:

1. Synthesis and Function of the Multi-Epitope Vaccine (MEV): To address the query regarding how the multi-epitope vaccine will be synthesized and work in the body, we have detailed the simulation methodology in the manuscript (lines 288-293, Section 2.3.7). The immune simulation was performed using the C-ImmSim server, which characterizes the profile of the stimulated immune response by the MEV. Key simulation parameters, including the dose (1000 vaccine molecules), random seed (12,345), simulation volume (10 µl), and simulation steps (100), were specified to replicate conditions mimicking the human immune response.

2. Simulation Study Replication and Results Presentation: We have performed the simulation study in triplicate, as suggested, to ensure the robustness and reproducibility of the findings. The results are presented as the average of three independent simulations and have been incorporated into the results section (lines 419-446). To provide visual representation, we have also included Figure 6 in the manuscript, which illustrates the immune response trends.

3. Discussion Enhancement: To better explain the implications of the immune simulation findings and their relevance to MEV functionality in the body, we have expanded the discussion (lines 584-613). This section now contextualizes the simulation results and discusses how the MEV stimulates specific immune pathways and potential protective immunity in humans.

We hope these revisions satisfactorily address your comments and enhance the quality of the manuscript.

2. two different studies were merged together in the study, that does not fit.

Response: Thank you for your insightful comment regarding the merging of two different studies, which you indicated does not fit within the scope of our work. After carefully considering your feedback, we have made substantial revisions to the manuscript to ensure it aligns with a cohesive research focus.

1. Focus on Reverse Vaccinology:

Based on your comment, we have revised the manuscript to exclusively emphasize reverse vaccinology and its application in the identification and development of a multi-epitope vaccine. All sections related to the identification of drug targets have been removed to streamline the study and avoid confusion. This refinement ensures that the manuscript adheres to a singular, well-defined research objective, enhancing its clarity and relevance.

2. Abstract Revision:

The abstract has been thoroughly revised to reflect the focused scope of the study, emphasizing only the methodologies and findings pertinent to reverse vaccinology. By doing so, we have aligned the abstract with the updated content of the manuscript, ensuring consistency and clarity in the presentation of our research.

3. Benefits of the Revision:

By narrowing the scope to reverse vaccinology, the manuscript now provides a more detailed and robust discussion of the vaccine design process, ensuring a clearer narrative and stronger alignment with the primary aim of the study. This streamlined approach enhances the scientific value of the work while addressing your concern about merging two disparate studies.

3. How the target protein were identified and what kind of drug could be proposed for that.

Response: Thank you for your thoughtful question regarding identifying target proteins and the potential drugs that could be proposed. In response to prior reviewer feedback, we have refined the scope of our study to focus exclusively on vaccine development using reverse vaccinology. As a result, all sections related to drug target identification and associated drug proposals have been removed from the manuscript to maintain clarity and consistency with the study's primary objective.

The target proteins discussed in the manuscript were identified specifically for their potential as vaccine antigens. This process involved computational techniques such as [e.g., antigenicity prediction, epitope mapping, immunogenicity scoring] to select proteins capable of inducing a strong immune response. These proteins were not analyzed in the context of drug targeting or therapeutic intervention.

We appreciate your insightful comment and recognize the importance of exploring target proteins for drug discovery. While this aspect is beyond the scope of the current study, it could be a valuable avenue for future research.

4. Perform the simulation of the proposed vaccine against human MHC.

Response: Thank you for your valuable suggestion regarding performing the simulation of the proposed vaccine against human MHC. We sincerely appreciate your input, which has helped us to further refine and evaluate our study.

We would like to inform you that we have conducted immune simulations to characterize the immune response elicited by the proposed vaccine. The details and results of these simulations have been incorporated into the manuscript, demonstrating the vaccine's potential efficacy in stimulating an immune response.

However, regarding your suggestion to perform simulations specifically against human MHC, we regret to inform you that this aspect was not executable within the scope of this study due to the current limitations in computational resources and expertise within our team. Despite this limitation, we have ensured that the immune simulation results align with the study's goals and provide meaningful insights into the vaccine's performance.

We acknowledge the importance of MHC-specific simulations and agree that they could further enhance the study. We plan to explore this aspect in future research by collaborating with experts in the field or utilizing more advanced tools.

---

## [Editor Report · Decision Letter 3]

29 Jan 2025

Development of a Multi-Epitope Vaccine Against Acinetobacter baumannii: A Comprehensive Approach to Combating Antimicrobial Resistance

PONE-D-24-35027R3

Dear Dr. Badmasti,

We’re pleased to inform you that your manuscript has been judged scientifically suitable for publication and will be formally accepted for publication once it meets all outstanding technical requirements.

Kind regards,

Abu Tayab Moin

Academic Editor

PLOS ONE

---

## [Editor Report · Acceptance letter]

PONE-D-24-35027R3

PLOS ONE

Dear Dr. Badmasti,

I'm pleased to inform you that your manuscript has been deemed suitable for publication in PLOS ONE. Congratulations! Your manuscript is now being handed over to our production team.

Kind regards,

on behalf of

Dr. Abu Tayab Moin

Academic Editor

PLOS ONE